# Towards Enhancing Relational Rules for Knowledge Graph Link Prediction

**Shuhan Wu**[1], **Huaiyu Wan**[1,3], **Wei Chen**[1], **Yuting Wu**[2*], **Junfeng Shen**[1] and **Youfang Lin**[1,3]

[1]School of Computer and Information Technology, Beijing Jiaotong University, Beijing, China
[2]School of Software Engineering, Beijing Jiaotong University, Beijing, China
[3]Beijing Key Laboratory of Traffic Data Analysis and Mining, Beijing, China
`{wushuhan, hywan, w_chen, ytwu1, jfshen, yflin}@bjtu.edu.cn`

## Abstract

Graph neural networks (GNNs) have shown promising performance for knowledge graph reasoning. A recent variant of GNN called *progressive relational graph neural network* (PRGNN), utilizes relational rules to infer missing knowledge in relational digraphs and achieves notable results. However, during reasoning with PRGNN, two important properties are often overlooked: (1) the **sequentiality of relation composition**, where the order of combining different relations affects the semantics of the relational rules, and (2) the **lagged entity information propagation**, where the transmission speed of required information lags behind the appearance speed of new entities. Ignoring these properties leads to incorrect relational rule learning and decreased reasoning accuracy. To address these issues, we propose a novel knowledge graph reasoning approach, the **R**elational r**U**le e**N**hanced **G**raph **N**eural **N**etwork (RUN-GNN). Specifically, RUN-GNN employs a *query related fusion gate unit* to model the sequentiality of relation composition and utilizes a *buffering update mechanism* to alleviate the negative effect of lagged entity information propagation, resulting in higher-quality relational rule learning. Experimental results on multiple datasets demonstrate the superiority of RUN-GNN is superior on both transductive and inductive link prediction tasks.

## 1 Introduction

Knowledge graph (KG) such as FreeBase (Bollacker et al., 2008), NELL (Carlson et al., 2010) and YAGO (Suchanek et al., 2007), is essentially a semantic network used to store and organize knowledge, which is extensively applied in a variety of scenarios, such as question answering (Yasunaga et al., 2021; Galkin et al., 2022), recommendation systems (Wang et al., 2018), semantic search (Berant and Liang, 2014), etc. Each of these KGs, how-ever, faces the problem of incompleteness, making it difficult to provide effective knowledge services for downstream applications. As a result, KG reasoning, known as link prediction in KG, is proposed to automatically complete the missing knowledge and has attracted much attention from researchers.

Various methods for link prediction are explored to facilitate reasoning for missing knowledge. Previous methods such as TransE (Bordes et al., 2013) and ConvE (Dettmers et al., 2018) perform reasoning by learning and utilizing distributed representations of entities and relations in triples. Since these methods can only capture the features of a single triple, some methods such as MINERVA (Das et al., 2018) and M-walk (Shen et al., 2018), are proposed to mine the path semantic information composed of triples. To further learn the semantic association of graph structures, methods such as CompGCN (Vashishth et al., 2019) and KE-GCN (Yu et al., 2021) use graph neural networks (GNNs) to aggregate neighbor information.

The recent *progressive relational graph neural network* (PRGNN) such as NBFNet (Zhu et al., 2021) and RED-GNN (Zhang and Yao, 2022), is a kind of advanced KG link prediction method that learns relational rules to infer missing knowledge. Relational rules refer to the Horn Clause that consists only of ordered relations, such as $has\_father\_in\_law(a, c) \leftarrow has\_wife(a, b) \land has\_father(b, c)$. In PRGNN-based methods, the representation of each entity encodes the information of the specific relational digraph (r-digraph) (Zhang and Yao, 2022). Yet, we observe that **the sequentiality of relation composition** and **lagged entity information propagation** would affect the quality of encoded relational rule during inference, and the aforementioned methods lack the ability to deal with these two properties.

**The sequentiality of relation composition** means that the order of combining differ-

---

*Corresponding author.

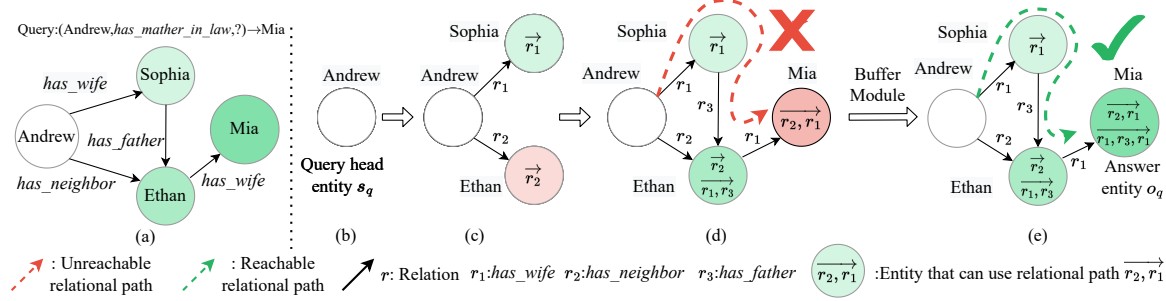

Figure 1: Figure (a) shows a toy KG. Figures (b), (c), and (d) depict the sequential inference process of the PRGNN-based methods for the query $(Andrew, has\_mather\_in\_law, ?) \rightarrow Mia$ on the toy KG. (e) is the state of the KG after using our buffer module. $\overrightarrow{r_2, r_1}$ refers to the relational path $r_2(a, b) \wedge r_1(b, c)$. The representations of entities depicted in Figure (c) are generated by employing the entity representations from Figure (b) and the relations that interconnect them, including Figures (d) and (e).

ent relations affects the semantics of the relational rules. For example, both the relational rules $has\_father(a, c) \leftarrow has\_sister(a, b) \wedge has\_father(b, c)$ and $has\_aunt(a, c) \leftarrow has\_father(a, b) \wedge has\_sister(b, c)$ contain the two same relations $has\_sister$ and $has\_father$. The combination of two different relations in a different order can lead to different conclusions. However, existing PRGNN-based methods face limitations in encoding such sequential information due to the standard practice of using addition and multiplication operations to integrate representations of entities and relations. Consequently, these methods are prone to misinterpreting the encoded relational rule information, which ultimately restricts their reasoning ability.

The phenomenon of **lagged entity information propagation** refers to the situation where the transmission speed of required information lags behind the appearance speed of new entities. The PRGNN updates entity representations by propagating information on a growing subgraph, with entities in the subgraph considered as candidate answer entities. Consequently, as PRGNN undergoes forward propagation, new candidate answer entities continually emerge. However, these new entities often possess limited relational path information, while a substantial amount of crucial information remains trapped within the old candidate answer entities without being promptly conveyed to the new entities. Taking Figure 1 as an example, suppose we want to infer who *Andrew*'s mother-in-law is. The correct answer entity *Mia* is a new candidate answer entity in Figure 1 (d) that can only obtain information about the shortest relational path $\overrightarrow{r_2, r_1}$ from *Andrew* to *Mia*. The true key information $\overrightarrow{r_1, r_3}$,

however, remains in the old candidate answer entity *Ethan*. Therefore, it becomes challenging to make the correct judgment for the given question. This phenomenon makes the model more prone to learning erroneous relation rules during the training phase and producing incorrect conclusions during the inference phase.

To address the properties mentioned above, we propose a novel KG link prediction method called **R**elational r**U**le e**N**hanced **G**raph **N**eural **N**etwork(RUN-GNN), equipped with two strategies for enhancing relational rules. The first enhancement strategy involves employing a *query related fusion gate unit* to update the representation of relational rules. This allows RUN-GNN to better encode various combination patterns among relations and identify the most valuable relational rules from a pool of candidate rules. The second proposed strategy is the utilization of a *buffering update mechanism*. This involves adding a buffer module to the PRGNN encoder, allowing the transmission of lagged information to the entities that require it. By doing so, the candidate answer entities promptly receive the necessary information, thus reducing the risk of the model learning incorrect rules. Both of the above strategies allow the method to learn and utilize enhanced relational rules effectively, thereby improving the performance of the method.

Briefly, the main contributions of this paper are as follows:

- To the best of our knowledge, we are the first to study the sequentiality of relation composition and the lagged entity information propagation in KG link prediction, and a novel PRGNN-based KG link prediction framework

RUN-GNN is proposed to enhance relational rules.

- A *query related fusion gate unit* (QRFGU) is proposed to orderly fuse different relational rules according to the query relation. Using QRFGU as a message-passing function can model the sequentiality of relation composition and significantly improve inference performance.

- A *buffering update mechanism* is designed to help entities obtain richer reasoning rule information and overcome the problem of the lagged entity information propagation.

- Experiments are conducted on multiple datasets under the inductive and transductive settings, and the experimental results show that RUN-GNN obtains substantial improvement over state-of-the-art methods.

## 2 Related Work

### 2.1 Inductive Knowledge Graph Reasoning

Inductive KG reasoning methods can perform reasoning on unseen entities. Some inductive methods aggregate the representations of seen entities to generate representations for unseen entities. Examples of such methods include LAN(Wang et al., 2019a) and CFAG (Wang et al., 2022). There are also some inductive methods that do not rely on representations of seen entities at all, instead relying solely on relational subgraphs for reasoning. Examples of these methods include GraIL (Teru et al., 2020) and CoMPILE (Mai et al., 2021), RED-GNN (Zhang and Yao, 2022).

### 2.2 Triple Information based Link Prediction

Methods based on triple information directly reason on triples with the entity and relation representations, including TransE (Bordes et al., 2013), TransR (Lin et al., 2015), TransH (Wang et al., 2014), HypE (Fatemi et al., 2021), RotatE (Sun et al., 2018), DistMult (Yang et al., 2015), ConvE (Dettmers et al., 2018), HAKE (Zhang et al., 2020), HousE (Li et al., 2022). These methods are simple and efficient, but cannot take advantage of graph structure features and are not interpretable.

### 2.3 Path Information based Link Prediction

Methods based on path information mainly include path-based methods and rule-based meth-

ods. Path-based methods predict triples by learning and utilizing paths, including MINERVA (Das et al., 2018), M-walk (Shen et al., 2018), CURL (Zhang et al., 2022) etc. The rule-based methods find rules by mining a series of paths and employ those paths with high reliability as rules for reasoning, including RNNLogic (Qu et al., 2020), DRUM (Sadeghian et al., 2019), NeuralLP (Yang et al., 2017), RLogic (Cheng et al., 2022). Path information based methods are well interpretable but challenging to reason with long paths.

### 2.4 Graph Structure Information based Link Prediction

#### 2.4.1 Graph Neural Network based Link Prediction

After the introduction of Graph Convolutional Networks (GCN) (Kipf and Welling, 2016) for homogeneous graphs, the attention of researchers was also drawn to the heterogeneous graph. R-GCN (Schlichtkrull et al., 2018), which focuses on the relations between entities, was quickly proposed. Subsequent researchers further developed various graph neural network methods for heterogeneous graph reasoning, such as HAN (Wang et al., 2019b), BA-GNN (Iyer et al., 2021), CompGCN (Vashishth et al., 2019) and KE-GCN (Yu et al., 2021).

#### 2.4.2 Subgraph-based Link Prediction

In contrast to conventional GNN-based approaches, subgraph-based methods often explicitly sample and encode neighborhood subgraphs of entities for reasoning. Early subgraph-based methods required sampling multi-hop subgraphs for each involved entity during each inference step, resulting in a high time complexity that limited their application to small datasets and relation prediction tasks. Examples of such methods include GraIL (Teru et al., 2020), SNRI (Xu et al., 2022), LogCo (Pan et al., 2022), ConGLR (Lin et al., 2022), CoMPILE (Mai et al., 2021), CFAG (Wang et al., 2022).

#### 2.4.3 PRGNN-based Link Prediction

The PRGNN-based reasoning method is an advanced subgraph-based KG link prediction method that performs inference by encoding r-digraph sequences as relational rule representation, including NBFNet (Zhu et al., 2021) and RED-GNN (Zhang and Yao, 2022). Zhu et al. (2021) extended the Bellman-Ford algorithm using neural networks to propose NBFNet, whose best-performing instance model follows the reasoning pattern of PRGNN,

providing the first validation of the effectiveness of PRGNN. Zhang and Yao (2022) formally proposed the efficient progressive relational graph neural network framework RED-GNN for the first time by recursively encoding the r-digraphs. This significantly improves the problem of high time complexity of subgraph-based methods.

## 2.5 Link Prediction with Extra Information

Most KG reasoning methods typically learn how to perform reasoning from existing factual triplets. However, some methods have also incorporated additional information for reasoning. For instance, methods like JOIE (Hao et al., 2019) and DGS (Iyer et al., 2022) introduce ontology information, while KG-BERT (Yao et al., 2019) and StAR (Wang et al., 2021) incorporate textual information. MKG-former (Chen et al., 2022), on the other hand, leverages extra multimodal information for reasoning.

## 3 Methodology

### 3.1 Problem Definition

The knowledge graph is denoted as $\mathcal{G} = (\mathcal{V}, \mathcal{R}, \mathcal{F})$, where $\mathcal{V}$ represents entity set, $\mathcal{R}$ represents relations set, and $\mathcal{F} = \{(s, r, o)|s, o \in \mathcal{V}, r \in \mathcal{R}\}$ represents the fact triples set.

The task of link prediction in KG aims to infer the missing entity $o_q$ by giving an incomplete query triple $(s_q, q, ?)$. To simplify the process of link prediction, we follow the works (Vashishth et al., 2019; Sadeghian et al., 2019) to augment the KG by adding inverse and identity triples.

Depending on whether the entities in the test set appear in the training set, link prediction tasks can be divided into inductive settings and transductive settings. With the inductive setting, entities in the test sets do not appear in the training set. Our proposed method RUN-GNN is capable of both transductive and inductive tasks.

### 3.2 Progressive Relational Graph Neural Network

PRGNN is an advanced variant of GNNs, which performs KG link prediction by learning relational rules in KG. Unlike traditional GNNs, which update the representations of all entities in the graph during each propagation, the PRGNN-based approach only updates the representations of $i$-hop neighbors of the query head entity during the $i$-th propagation. In addition, it does not learn representations of entities, but only learns representations

of relations and relational rules. It encodes the r-digraph into a rule information representation for link prediction.

Both rule-based methods and PRGNN-based methods use only the combinations of relations for reasoning, but there are some differences in the form of utilization. Rule-based methods search for candidate answer entities in the neighborhood of the query head entity using a set of relational rules and select the answer entity from the candidate answer entities based on the matching rules. In contrast, PRGNN-based methods consider all entities in the neighborhood as candidate answer entities and encode the r-digraph as representations for the candidate entities using GNNs. The answer entity is then selected based on the representations of the entities. Therefore, PRGNN-based methods perform inference by learning relational rules, in which the entity representations can be considered representations of relational rules.

### 3.3 Model Architecture

Figure 2 illustrates the overall structure of our proposed method RUN-GNN. The RUN-GNN follows an encoder-decoder structure. The encoder contains an exploration module and a buffer module connected in series.

The primary objective of the exploration module is to explore more candidate answer entities and to generate entity representations of relational rules for them. The purpose of the buffer module, which operates as the latter component, is to update the information of changed relational rules to the relevant entities in a timely manner. The decoder is a linear layer that provides candidate answer entities with scores.

### 3.4 Query Related Fusion Gate Unit

The *query related fusion gate unit* (QRFGU), a variant of GRU(Chung et al., 2014), is designed to effectively model the properties of relational rules, specifically to ensure the sequentiality of relation composition. The QRFGU integrates the relation message representation $h_{msg} \in \mathbb{R}^d$ into the entity's rule representation $h_{rule} \in \mathbb{R}^d$ to obtain a new query-related relational rule representation $h_{fuse} \in \mathbb{R}^d$, where $d$ denotes the size of the embedding dimension.

The structure of QRFGU is shown in Figure 3. QRFGU can be denoted as:

$$h_{fuse} = QRFGU(h_{rule}, h_{msg}, h_q), \quad (1)$$

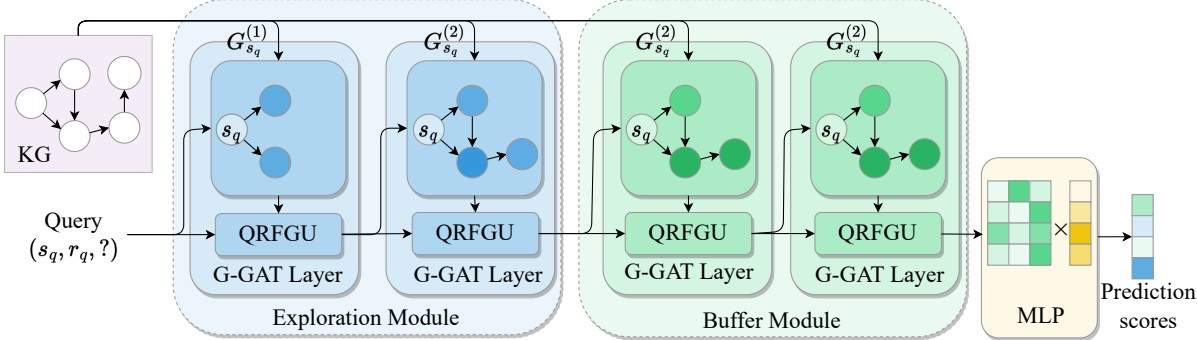

Figure 2: The overall structure of RUN-GNN. QRFGU: *query related fusion gate unit*. $G_{s_q}^{(i)}$: the $i$ hop subgraph centered on $s_q$.

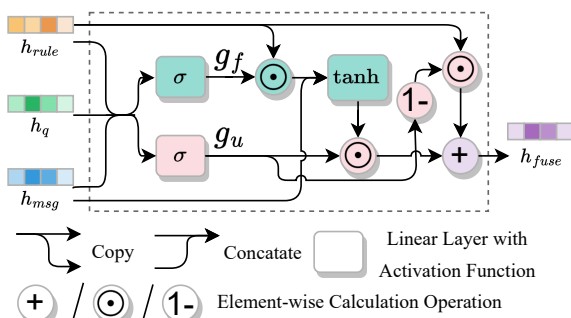

Figure 3: The structure of *query related fusion gate unit* (QRFGU).

where $h_q \in \mathbb{R}^d$ denotes query relation representation. The QRFGU computes the forget gate $g_f \in [0,1]^d$ and update gate $g_u \in [0,1]^d$ based on $h_{rule}$, $h_{msg}$, and $h_q$ firstly. And then, the QRFGU computes the candidate hidden state $h_c \in \mathbb{R}^d$ based on $g_f$, $h_{rule}$, and $h_{msg}$, and subsequently derives the fused state $h_{fuse}$ by combining $h_c$, $h_{rule}$, and $g_u$. The detailed calculation process of QRFGU can be described as:

$$g_u = \sigma(W_u[h_{rule}, h_{msg}, h_q] + b_u), \qquad (2)$$

$$g_f = \sigma(W_f[h_{rule}, h_{msg}, h_q] + b_f), \qquad (3)$$

$$h_c = \tanh(W_c(h_{msg} + (h_f \odot h_{rule})) + b_c), \ (4)$$

$$h_{fuse} = (1 - f_u) \odot h_{rule} + f_u \odot h_c. \qquad (5)$$

Among them, $\sigma$ represents the sigmoid activate function. $[,]$ denotes the concatenation operation. $W_u, W_f \in \mathbb{R}^{d \times 3d}, W_c \in \mathbb{R}^{d \times d}, b_u, b_f, b_c \in \mathbb{R}^d$ all denote trainable parameters, $\odot$ represents the Hadamard product.

## 3.5 Enhancing Relational Rules Using Query Related Fusion Gate Unit

In RUN-GNN , the exploration module consists of $n$ Gated Graph Attention Network (G-GAT) layers,

in which the $i$-th G-GAT layer propagates relational information in $i$-hop subgraphs $G_{s_q}^{(i)}$ centered on the query head entity $s_q$, treating entities in these subgraphs as candidate answer entities and generating corresponding entity representations of relational rules.

To better handle the sequentiality of relation composition, the G-GAT layer leverages the QRFGU to fuse the representations of the head entity and relation of each triple $(s, r, o)$ and generates the representation of the corresponding candidate relation rule message $m_{s,r,o,q}$. The computation of $m_{s,r,o,q}$ is formalized as follows:

$$m_{s,r,o,q}^l = QRFGU(h_s^{l-1}, h_r^l, h_q^l), \qquad (6)$$

where $h_s, h_r \in \mathbb{R}^d$ represent the representation of the head entity and relation of the current triple separately, the variable $l$ in $h_r^l$ denotes the $l$-th G-GAT layer.

The G-GAT layers compute the attention score $a_{s,r,o,q}^l$ for each triple $(s, r, o)$ in the subgraph $G_{s_q}^l$ based on $h_q^l$, $m_{s,r,o,q}^l$. The attention score $a_{s,r,o,q}^l$ is calculated by:

$$c_{s,r,o,q}^l = W_a^l ReLU(W_s^l m_{s,r,o,q}^l + W_q^l h_q^l), \quad (7)$$

$$a_{s,r,o,q}^l = \frac{\exp(c_{s,r,o,q}^l)}{\sum_{(\tilde{s},\tilde{r},o) \in \mathcal{T}_o^l} \exp(c_{\tilde{s},\tilde{r},o,q}^l)}, \qquad (8)$$

where $W_a^l \in \mathbb{R}^{1 \times d_a}$ and $W_s^l, W_q^l \in \mathbb{R}^{d_a \times d}$ represent trainable weight matrices, $d_a$ is the attention space dimension. $\mathcal{T}_o^l$ is the set of triples centered on $o$ in $G_{s_q}^{(l)}$. So the candidate relational rule representations for entities are calculated by the Equation:

$$\tilde{h}_o^l = \varphi\left(\sum_{(s,r,o) \in \mathcal{T}_o^l} a_{s,r,o,q}^l m_{s,r,o,q}^l\right), \qquad (9)$$

where $\varphi$ represents the $ReLU$ activation function, $tanh$ activation function or no operation.

After that, to further retain the valuable existing relational rules, RUN-GNN utilizes QRFGU to control the updating of representations of entities. The $l$-th G-GAT layer calculates the representation of any entity $o$ in $G_{s_q}^{(i)}$ in by:

$$h_o^l = QRFGU(h_o^{l-1}, \tilde{h}_o^l, h_q). \qquad (10)$$

It is worth emphasizing that the exploration module commences by initializing all entity representations as zero vectors.

## 3.6 Enhancing Relational Rules Using Buffering Update Mechanism

The generation of representations for candidate answer entities is a key feature of the exploration module. However, incomplete encoding of relational rule information presents a challenge due to lagged entity information propagation. A possible solution is to increase the number of G-GAT layers in the exploration module. However, this approach may lead to a larger subgraph during inference, resulting in higher resource consumption and a larger number of candidate answer entities. Furthermore, new candidate answer entities are still susceptible to lagged entity information propagation.

In this work, we propose a simple and elegant *buffering update mechanism* that can alleviate the problem of lagged entity information propagation with relatively low resource consumption, and significantly improve the performance of the model in queries where the answer entity is far away. In detail, We add a buffer module after the exploration module, which consists of $m$ G-GAT layers, each of which propagates information on the same subgraph $G_{s_q}^{(n)}$. The buffer module acts as a buffer that allows candidate answer entities to wait for important relational rule information retained in dependent entities. This ensures that all candidate answer entities have enough relational rule information before proceeding with decoding.

## 3.7 Model Prediction and Optimization

The score of each entity is calculated based on a representation containing the appropriate relational rules. The score of entity $o$ is given by

$$score(o) = f(s_q, q, o) = W_{score} h_o^{n+m}, \qquad (11)$$

where $W_{score} \in \mathbb{R}^{1 \times d}$ is the weight matrix. If an entity in RUN-GNN doesn't obtain a representation

from encoder at last, it will be assigned a score that values 0.

The link prediction task can be regarded as a multi-label classification problem. To optimize the parameters of the model, we use the multi-class log-loss (Lacroix et al., 2018; Zhang and Yao, 2022), i.e.

$$L = \sum_{(s_q, q, o_q) \in \mathcal{F}_{train}} (-f(s_q, q, o_q) + log(\sum_{o \in \mathcal{V}} \exp(f(s_q, q, o)))),$$
$$(12)$$

where $(s_q, q, o_q)$ represents positive triple in train triples $\mathcal{F}_{train}$, and $(s_q, q, o)$ denotes any triple with the same query $(s_q, q, ?)$.

# 4 Experiments

To demonstrate the effectiveness of RUN-GNN, many experiments under inductive and transductive tasks are conducted for KG link prediction on multiple benchmark datasets.

## 4.1 Transductive Experiments

### 4.1.1 Datasets and Baselines

We use four commonly used public datasets to conduct our experiments, including WN18RR (Dettmers et al., 2018), FB15k-237 (Toutanova and Chen, 2015), NELL-995 (Xiong et al., 2017) and YAGO3-10 (Chami et al., 2020). The details of these datasets can be found in Appendix B.1.

To verify the effectiveness of our method, we compare RUN-GNN with various types of KG link prediction methods, including triple information based methods ConvE (Dettmers et al., 2018), HousE (Li et al., 2022), HAKE (Zhang et al., 2020) and RotatE (Sun et al., 2018); path information based methods MINERVA (Das et al., 2018), DRUM (Sadeghian et al., 2019), CURL (Zhang et al., 2022) and RNNLogic (Qu et al., 2020); normal GNN-based methods CompGCN (Vashishth et al., 2019); existing PRGNN-based methods RED-GNN (Zhang and Yao, 2022) and NBFNet (Zhu et al., 2021). For most baseline methods, the experimental results are derived from related published papers and (Zhang and Yao, 2022). Since there are problems with the evaluation methodology of the RED-GNN, we re-evaluate the performance of the method using the code and hyperparameters published in paper (Zhang and Yao, 2022). We also evaluate the performance of CURL, RNN-Logic, CompGCN, RED-GNN, and NBFNet on YAGO3-10 using publicly available codes and hyperparameters.

| Type | Methods | WN18RR | | | FB15k-237 | | | NELL-995 | | | YAGO3-10 | | |
|---|---|---|---|---|---|---|---|---|---|---|---|---|---|
| | | MRR | Hit@1 | Hit@10 | MRR | Hit@1 | Hit@10 | MRR | Hit@1 | Hit@10 | MRR | Hit@1 | Hit@10 |
| Triple | ConvE | 0.430 | 39.0 | 49.0 | 0.325 | 23.7 | 50.1 | 0.514 | 44.2 | 63.2 | 0.440 | 35.0 | 62.0 |
| | QuatE | 0.480 | 44.0 | 55.1 | 0.350 | 25.6 | 53.8 | 0.533 | 46.6 | 64.3 | 0.495 | 40.2 | 67.0 |
| | HousE | 0.511 | 46.5 | 60.2 | 0.361 | 26.6 | 55.1 | 0.528 | 45.8 | 64.5 | _0.571_ | _49.1_ | _71.4_ |
| | HAKE | 0.497 | 45.2 | 58.2 | 0.346 | 25 | 54.2 | 0.527 | 45.9 | 64.1 | 0.545 | 46.2 | 69.4 |
| Path | MINERVA | 0.448 | 41.3 | 51.3 | 0.293 | 21.7 | 45.6 | 0.513 | 41.3 | 63.7 | - | - | - |
| | DRUM | 0.486 | 42.5 | 58.6 | 0.343 | 25.5 | 51.6 | 0.365 | 30.0 | 48.8 | - | - | - |
| | CURL | 0.462 | 42.9 | 52.7 | 0.2902 | 21.1 | 45.3 | 0.442 | 32.8 | 56.4 | 0.499 | 42.2 | 63.9 |
| | RNNLogic | 0.483 | 44.6 | 55.8 | 0.344 | 25.2 | 53.0 | 0.479 | 43.1 | 57.1 | 0.536 | 47.1 | 64.2 |
| Graph | CompGCN | 0.479 | 44.3 | 54.6 | 0.355 | 26.4 | 53.5 | 0.518 | 44.6 | 63.4 | 0.354 | 25.7 | 54.4 |
| | RED-GNN | 0.547 | _50.1_ | 63.5 | 0.376 | 28.2 | 56.0 | _0.548_ | _48.3_ | _65.7_ | 0.520 | 44.2 | 66.1 |
| | NBFNet | _0.551_ | 49.7 | _66.6_ | _0.415_ | **32.1** | _59.9_ | 0.501 | 41.1 | 65.6 | 0.373 | 27.9 | 56.9 |
| | **RUN-GNN (ours)** | **0.586** | **53.2** | **68.8** | **0.416** | _31.9_ | **61.0** | **0.580** | **51.6** | **68.4** | **0.580** | **50.5** | **71.5** |

Table 1: Experimental results on WN18RR, FB15k-237, NELL-995, YAGO3-10 under transductive settings. Hit@N values are in percentage. The best scores are in bold and the second-best scores are with underline. '-' means unavailable results.

### 4.1.2 Detailed Settings

To evaluate our proposed method's performance, we utilize the filtered Mean Reciprocal Rank (MRR), Hit@1, and Hit@10 metrics. Our method is implemented using PyTorch (Paszke et al., 2019) and PyG (Fey and Lenssen, 2019).[1] To train the model, we utilize four NVIDIA RTX A4000 GPUs for 80 epochs. The best performance of models is selected based on the MRR metric on each validation set. Further details on the transductive experimental setup, including experimental hyperparameters, training time, number of model parameters, and more, can be found in Appendix B.2.

### 4.1.3 Results and Discussion

Table 1 illustrates the experiment results of different methods. Our RUN-GNN model outperforms baselines based on triple and path information on all datasets by a significant margin. RUN-GNN also achieves significant performance improvements over other PRGNN-based methods. For instance, RUN-GNN achieves an improvement of 6.35% over NBFNet for the MRR metric on the WN18RR dataset. The results show our method currently has the most powerful reasoning ability.

Our proposed method, RUN-GNN , along with other methods like NBFNet and REDGNN, utilize PRGNN reasoning mode, which gives RUN-GNN a significant advantage over traditional reasoning methods. Compared to ConvE, RotatE, and MINERVA, RUN-GNN can utilize more graph structural information in KG. Also, RUN-GNN can flexibly employ complex rules for link prediction implicitly, unlike rule-based methods like DRUM and

| | WN18RR | | | | FB15k-237 | | | |
|---|---|---|---|---|---|---|---|---|
| Methods | V1 | V2 | V3 | V4 | V1 | V2 | V3 | V4 |
| Neural LP | 0.649 | 0.635 | 0.361 | 0.628 | 0.325 | 0.389 | 0.400 | 0.396 |
| DRUM | 0.666 | 0.646 | 0.380 | 0.627 | 0.333 | 0.395 | 0.402 | 0.410 |
| GraIL | 0.627 | 0.625 | 0.323 | 0.553 | 0.279 | 0.276 | 0.251 | 0.227 |
| RED-GNN | _0.693_ | _0.687_ | _0.422_ | _0.642_ | _0.341_ | _0.411_ | _0.411_ | _0.421_ |
| NBFNet | 0.686 | 0.662 | 0.410 | 0.601 | 0.270 | 0.321 | 0.335 | 0.288 |
| **RUN-GNN** | **0.699** | **0.697** | **0.445** | **0.654** | **0.397** | **0.473** | **0.468** | **0.463** |

Table 2: Experimental results under inductive settings (evaluated with MRR). The best score is in bold and the second-best scores are with underline.

RNNLogic. Compared to CompGCN, RUN-GNN can represent every possible answer entity related to the current query.

Compared to NBFNet and REDGNN, our proposed method, RUN-GNN , attributes PRGNN's reasoning capability to learning relational rules, and employs two effective strategies to address critical properties affecting PRGNN's reasoning capability: sequential relation composition and lagged entity information propagation. These strategies enhance RUN-GNN 's ability to encode relational rules and give it a clear advantage over similar methods.

### 4.2 Inductive Experiments

We conduct inductive experiments to evaluate methods' ability to reason about unseen entities.

### 4.2.1 Datasets and Baselines

We select eight widely used inductive datasets derived from WN18RR and FB15k-237 following the settings of previous research (Zhang and Yao, 2022; Teru et al., 2020).

In this work, we compare RUN-GNN with several other methods that possess inductive link prediction ability, including Neural LP (Yang et al.,

---

[1]Code is available at https://github.com/Ninggirsu/RUN-GNN.

2017), DRUM (Sadeghian et al., 2019), GraIL (Teru et al., 2020), RED-GNN (Zhang and Yao, 2022), NBFNet (Zhu et al., 2021). Among them, the experimental results of NeuralLP, DRUM, and GraIL are from the paper (Zhang and Yao, 2022). We re-evaluate the experimental results of RED-GNN and NBFNet using the code and hyperparameters published in their papers (Zhang and Yao, 2022; Zhu et al., 2021) due to issues with the experimental settings.

We evaluate the performance of the methods using the filtered MRR metric. We select the best performance based on the MRR metric on each relevant validation set. The detailed settings are provided in Appendix B.5.

### 4.2.2 Results and Discussion

Based on the results presented in Table 2, our model RUN-GNN performs well on both sub-datasets of WN18RR and FB15k-237. NeuralLP and DRUM use the mined chain-like rules to predict knowledge, so they are able to achieve good results. GraIL samples subgraphs between head and tail entities of triples and mines structural and relational information for reasoning, but doesn't effectively use intermediate relational rules. PRGNN-based methods, like RED-GNN and NBFNet, are able to encode complex relational rules into entity representations and thus perform well. RUN-GNN utilizes QRFGU to encode more complex rules with greater accuracy, which enhances the model's ability to leverage relational rules and leads to better performance.

### 4.3 Analysis

We conduct several experiments in this subsection to validate the effectiveness of our proposed method. For subsequent experiments, we utilize only the transductive setting since it is the most commonly used experimental setting (Vashishth et al., 2019; Yang et al., 2015; Bordes et al., 2013). All methods are tested using the WN18RR dataset, with $n$ set to 5, $m$ set to 3, and $d$ set to 64.

### 4.3.1 Ablation Study

We design several variants of RUN-GNN to evaluate the impact of each component. Table 3 presents the results of the experiments. The table shows that variant w/ multiplication and variant w/ addition, which use multiplication or addition to fuse the entity and relation representation, perform worse than RUN-GNN, which indicates that QRFGU is an ef-

| Methods | MRR | Hit@1 | Hit@3 | Hit@10 |
|---|---|---|---|---|
| w/ multiplication | 0.552 | 50.5 | 57.7 | 64.3 |
| w/ addition | 0.561 | 51.2 | 58.5 | 65.2 |
| w/o buffer | 0.563 | 51.4 | 58.8 | 65.5 |
| RUN-GNN | **0.571** | **52.5** | **59.2** | **66.1** |

Table 3: Experiment results of different variants.

| $n$ | $m$ | 1-Hop | 2-Hop | 3-Hop | 4-Hop | 5-Hop | 6-Hop | ALL |
|---|---|---|---|---|---|---|---|---|
| 2 | 0 | 0.994 | 0.449 | 0.001 | 0.000 | 0.001 | 0.000 | 0.391 |
| 2 | 2 | 0.995 | 0.496 | 0.001 | 0.001 | 0.001 | 0.000 | 0.394 |
| 3 | 0 | 0.995 | 0.533 | 0.383 | 0.000 | 0.001 | 0.000 | 0.480 |
| 3 | 2 | 0.995 | 0.544 | 0.452 | 0.000 | 0.001 | 0.000 | 0.496 |
| 4 | 0 | 0.996 | 0.602 | 0.516 | 0.103 | 0.000 | 0.000 | 0.524 |
| 4 | 2 | 0.996 | 0.604 | 0.544 | 0.155 | 0.000 | 0.000 | 0.533 |
| 5 | 0 | 0.996 | 0.644 | 0.612 | 0.212 | 0.101 | 0.000 | 0.565 |
| 5 | 2 | 0.996 | 0.626 | 0.621 | 0.244 | 0.125 | 0.000 | 0.570 |
| 6 | 0 | 0.996 | 0.628 | 0.614 | 0.282 | 0.143 | 0.029 | 0.574 |
| 6 | 2 | 0.996 | 0.617 | 0.621 | 0.298 | 0.168 | 0.039 | 0.579 |
| Count | | 2192 | 582 | 1346 | 470 | 556 | 276 | 6268 |
| Ratio | | 34.97 | 9.29 | 21.47 | 7.50 | 8.87 | 4.40 | 100.00 |

Table 4: The performance comparison of the MRR between RUN-GNN models with different hyperparameters on the WN18RR dataset.

fective component for encoding relational rules. Additionally, variant w/o buffer, which does not use the *buffering update mechanism*, also performs worse than RUN-GNN, demonstrating how the buffer module can improve entity representation and link prediction.

### 4.3.2 Does Buffer Update Mechanism Solve Lagged Entity Propagation?

To evaluate the effectiveness of the proposed *Buffer Update Mechanism* in mitigating the negative effects of Lagged Entity Propagation, we classify the test set based on the shortest path length from the head to tail entity of the query triple, varying hyperparameters $m$ and $n$ to assess RUN-GNN 's link prediction performance. Results presented in Table 4 show that increasing the value of $m$ consistently improves the model's inference performance, particularly for triples with longer inference paths. This demonstrates the effectiveness of our proposed method in mitigating the negative effects of lagged entity information propagation. Additionally, results also reveal that the number of G-GAT layers in the model's exploration module significantly affects performance.

### 4.3.3 Does Query Related Fusion Gate Unit Deal Well with Sequentiality of Relation Composition?

We designed an experiment to validate the effectiveness of QRFGU in encoding the sequentiality

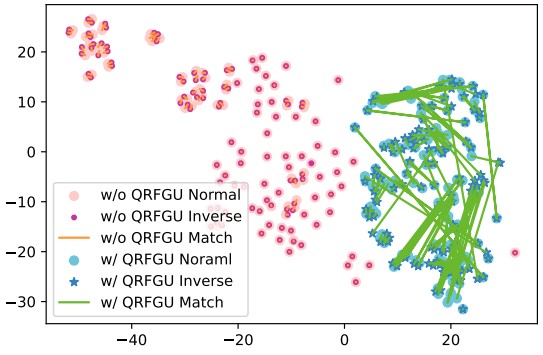

Figure 4: Similarity image representing relational paths encoded in different ways.

of relation composition.

Firstly, We selected lots of common relational paths from the WN18RR dataset and reversed the order of relations in these paths to obtain a set of reversed relational paths. Then these relational paths were separately encoded using the original RUN-GNN (represented as w/ QRFGU) and a variant that uses element-wise addition as the MESSAGE function (represented as w/o QRFGU). This resulted in four sets of representations. We mapped these representations to a 2-dimensional space and plotted the representations using different colored dots in Figure 4. We also connected the representations of the reversed relational paths and their corresponding original relational paths using colored lines.

In Figure 4, there are no distinct orange lines, indicating that the representations of different order combinations of the same relation generated by the variant w/o QRFGU are very close. The model fails to capture the sequentiality of compositional relations and cannot distinguish between these representations. On the other hand, there are clear green lines in Figure 4, indicating that the representations generated by RUN-GNN (w/ QRFGU) have significant differences. The model successfully captures the sequentiality of compositional relations and can distinguish between these representations.

### 4.4 The Time Complexity Analysis

The overall time complexity of RUN-GNN, which consists of $l$ GNN layers in the exploration module and $n$ GNN layers in the buffer module, can be expressed as $O((\min(nD^l, (l+n)|\mathcal{F}|) + |\mathcal{V}|)d^2)$, where $d$ is the dimension of relation representations in the model, $D = \frac{|\mathcal{F}|}{|\mathcal{V}|}$ is the average degree of the knowledge graph , $|\mathcal{V}|$ is the number of entities,

and $|\mathcal{F}|$ is the number of fact triples. More details can be found in the Appendix D.

## 5 Conclusion

This paper attributes PRGNN's reasoning capability to learning relational rules and identifies two issues with existing methods. Firstly, we identify the issue of ignoring the sequentiality of relation composition, which causes the model to become confused with different relational rules. To address this issue, we propose QRFGU. Additionally, we identify the issue of lagged entity information propagation, which can lead to erroneous rule learning. A *buffering update mechanism* is introduced to mitigate this problem. We then combine these two relation rule enhancement strategies to propose RUN-GNN , which achieves state-of-the-art performance on knowledge graph link prediction tasks. In the future, we plan to investigate more efficient link prediction methods based on PRGNN to improve their practicality.

## Limitations

There may be some possible limitations in this study.

(1) According to the work of Zhang and Yao (2022), the time complexity of the PRGNN inference mode followed by RUN-GNN is higher than conventional methods. As a result, the computational resource consumption of RUN-GNN is also higher compared to these conventional methods. Especially, Using QRFGU and *buffering update mechanism* may increase the computational cost during model inference. Since QRFGU is essentially a small neural network, rather than a parameter-free addition or multiplication operation, using this method may increase the computational cost. Additionally, the *buffering update mechanism* results in additional calculations through the use of GNN, which may also contribute to a rise in computational cost. More details about time complexity can be found in the Appendix D.

Despite these limitations, RUN-GNN is an inductive reasoning method that does not require significant resources to retrain the model when new entities are added or when facts in the knowledge graph change, unlike traditional methods. Additionally, we believe that pruning the model after training to reduce the number of candidate entities is a potential future research direction to improve efficiency and reduce resource consumption during

model inference.

(2) Our method may have difficulty in reasoning about query triples where the head and tail entities are far apart. Since RUN-GNN can only consider the $n$-hop neighbors of the query head entity as candidate answer entities and other entities will be considered unlikely to be correct answer entities. Increasing the number of G-GAT layers in the exploration module by $n$ can improve this, but this will come with a sharp increase in computational resource requirements, and Section 4.3.2 also shows that the effect of doing so may be poor. Therefore, it is difficult to provide correct reasoning for triples where the answer entity is far from the query head entity.

We believe that this issue can be addressed by allowing the model to conditionally select candidate answer entities, which can reduce the answer search space and improve the model's ability to learn longer relational rules.

## Ethics Statement

This study is conducted with full compliance with the ethical code set out in the ACL Code of Ethics.

The datasets used in our research are publicly available datasets previously constructed by other researchers. The content of these datasets is sourced from publicly available online knowledge bases (Bollacker et al., 2008), publicly available cognitive linguistics-based English dictionaries (Dettmers et al., 2018; Miller, 1995), and publicly available semantic machine learning systems (Carlson et al., 2010; Xiong et al., 2017). Other than that, all entities involved in the experiments were anonymized. Other than that, all entities involved in the experiments were anonymized.

RUN-GNN's utility extends to the completion and verification of existing knowledge graphs, such as Wikidata. However, like other knowledge graph link prediction methods, the information predicted by our method may be poisonous, biased, or wrong, so an additional safety review of the prediction results may be required. Furthermore, the potential misuse of this tool to predict private information from public sources necessitates caution.

## Acknowledgements

This work is supported by the National Key R&D Program of China (No. 2021QY1502) and the Talent Fund of Beijing Jiaotong University (No. 2023XKRC032).

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

|         | WN18RR | FB15k237 | NELL995 | YAGO3-10 |
|---------|--------|----------|---------|----------|
| Relation | 11     | 237      | 200     | 37       |
| Entity  | 40943  | 14541    | 74,536  | 123182   |
| Train   | 86835  | 272115   | 149678  | 1079040  |
| Valid   | 3034   | 17535    | 543     | 5000     |
| Test    | 3134   | 20466    | 2818    | 5000     |

Table 5: Datasets statistics in transductive link prediction experiments.

## A Complete Algorithm for the Encoder of RUN-GNN

The complete algorithm for the encoder of RUN-GNN is described in Algorithm 1.

---
**Algorithm 1** The encoder of RUN-GNN

---
**Input** query head entity $s_q$, query relation $q$, the number $n$ of exploration modules and the number $m$ of the auxiliary modules.

**Output** $H = \{h_e | e \in \mathcal{E}^n\}$.

1: initial $h_e^0 = 0$ where $e \in \mathcal{V}$, entity set $\mathcal{E}^0 = \{s_q\}$, triple set $\mathcal{T}^0 = \emptyset$, $i = 1$, $j = 1$.
2: **for** $i <= n$ **do**
3: $\quad \mathcal{T}^i = \{(s, r, o)|s \in \mathcal{E}^{i-1} \wedge (s, r, o) \in \mathcal{F}\}$, $\mathcal{E}^i = \{o|s \in \mathcal{E}^{i-1} \wedge (s, r, o) \in \mathcal{F}\} \cup \mathcal{E}^{i-1}$.
4: $\quad$ **for all** $e \in \mathcal{E}^i$ **do**
5: $\quad\quad$ update $h_e$ by Equation (6) - (10)
6: $\quad$ **end for**
7: $\quad i+ = 1$.
8: **end for**
9: **for** $j <= m$ **do**
10: $\quad$ **for all** $e \in \mathcal{E}^n$ **do**
11: $\quad\quad$ update $h_e$ by Equation (6) - (10).
12: $\quad$ **end for**
13: $\quad j+ = 1$.
14: **end for**
15: **return** $H = \{h_e | e \in \mathcal{E}^n\}$.

---

## B Experiment Details

### B.1 Transductive Datasets Statistics

In the transductive experiments, we mainly used four publicly available datasets: FB15k-237 (Toutanova and Chen, 2015), NELL-995 (Xiong et al., 2017), YAGO3-10 (Chami et al., 2020) and WN18RR (Dettmers et al., 2018). The statistical information of these datasets is shown in Table 5.

### B.2 Transductive Experiment Detailed Settings

For the WN18RR, FB15k37, YAGO3-10, and NELL-995 datasets, we empirically select the num-

| Datasets | | WN18RR | | | | FB15k-237 | | | |
|---|---|---|---|---|---|---|---|---|---|
| | | Entity | Relation | Fact | Prediction | Entity | Relation | Fact | Prediction |
| v1 | train | 2746 | 9 | 5410 | 1268 | 1594 | 180 | 4245 | 981 |
| | test | 922 | 9 | 1618 | 373 | 1093 | 180 | 1993 | 411 |
| v2 | train | 6954 | 10 | 15262 | 3706 | 1608 | 200 | 9739 | 2346 |
| | test | 5084 | 10 | 4011 | 852 | 1660 | 200 | 4145 | 947 |
| v3 | train | 12078 | 11 | 25901 | 6249 | 3668 | 215 | 17986 | 4408 |
| | test | 5084 | 11 | 6327 | 1140 | 2501 | 215 | 7406 | 1731 |
| v4 | train | 3861 | 9 | 7940 | 1902 | 4707 | 219 | 27203 | 6713 |
| | test | 7084 | 9 | 12334 | 2823 | 3051 | 219 | 11714 | 2840 |

Table 6: Datasets statistics in inductive link prediction experiments.

| Methods | Family | | | UMLS | | |
|---|---|---|---|---|---|---|
| | MRR | Hit@1 | Hit@10 | MRR | Hit@1 | Hit@10 |
| MINERVA | 0.885 | 82.5 | 96.1 | 0.825 | 72.8 | 96.8 |
| DRUM | 0.934 | 88.1 | 99.6 | 0.813 | 67.4 | 97.6 |
| CompGCN | 0.933 | 88.3 | 99.1 | 0.927 | 86.7 | 99.4 |
| REDGNN | **0.992** | **98.8** | **99.7** | 0.964 | 94.6 | 99.0 |
| RUN-GNN | 0.989 | **98.8** | 99.1 | **0.986** | **98.0** | **99.5** |

Table 7: Experimental results on Family and UMLS under transductive settings. Hit@N values are in percentage. The best scores are in bold and the second-best scores are with underline.

ber $n$ of G-GAT layers in the exploration module to be 8, 6, 4, and 6, respectively, with a dimension size $d$ set to 64, 48, 32, and 48, respectively. Additionally, we set the G-GAT layer number $m$ in the buffer module to 2 for YAGO3-10 and 3 for RUN-GNN on other datasets. We use four NVIDIA RTX A4000 GPUs for training for up to 80 epochs, with early stopping based on MRR on the valid dataset. The training time for one epoch using these hyperparameters is 42, 117, and 370 minutes for WN18RR, NELL-995, and FB15k-237, respectively. The best performance of models is selected based on the MRR metric on each valid dataset. Additionally, the RUN-GNN model employed in link prediction across the WN18RR, FB15k-237, and NELL-995 datasets utilizes 231k, 196k, and 223k parameters, respectively.

## B.3 Extra Transductive Experiment results

In addition to classic datasets like WN18RR, we also conducted experiments on other datasets. Table 7 presents the performance of our method on the Family dataset (Kok and Domingos, 2007) and UMLS dataset(Kok and Domingos, 2007). On the Family dataset, our method performs similarly to the state-of-the-art RED-GNN. However, on the UMLS dataset, our method clearly outperforms

other approaches.

## B.4 Inductive Datasets Statistics

In our inductive experiments, we follow the experimental settings of the previous researchers' work (Teru et al., 2020; Zhu et al., 2021) paper and select eight inductive datasets from the WN18RR (Dettmers et al., 2018) and FB15k-237 (Toutanova and Chen, 2015) datasets. However, there are some differences in the experimental settings of inductive tasks using inductive datasets across different works. To accurately assess our model's performance, we employ these datasets for link prediction tasks in line with the experimental settings of the RED-GNN (Zhang and Yao, 2022) paper. Each inductive dataset comprises distinct training and testing subsets, each consisting of "fact triples" and "predict triples." The inductive methods infer missing entities in the "prediction triples" based on "fact triples" in the set. Notably, the training and testing subsets are entirely separate and utilize the same set of relations but feature different entities. To provide a comprehensive overview, we present the statistical information of the inductive datasets in Table 6.

## B.5 Inductive Experiment Detailed Settings

We evaluate the performance of the methods using the filtered MRR metric. For RUN-GNN, we perform hyperparameter tuning of the number $n$ of G-GAT in the exploration module, choosing from $\{4, 5, 6, 7\}$, the number $m$, choosing from $\{0, 1, 2, 3, 4\}$, and $d$, selecting from $\{32, 48, 64\}$. Hyperparameter tuning is conducted for 24 hours using the Evolution Algorithm (Real et al., 2017). The model is trained on a single NVIDIA RTX A4000 GPU for 50 epochs. We select the best performance based on the MRR metric on each

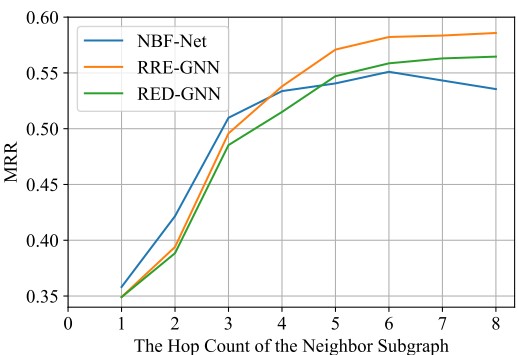

Figure 5: Comparison of inference capabilities of different PRGNN-based methods for message propagation on subgraphs of different sizes.

relevant validation set.

## C  The Influence of Information Propagation Range in PRGNN

The performance of PRGNN-based methods is highly dependent on the maximum size of subgraphs that GNN layers can propagate information through. This size is determined by the number $n$ of G-GAT Layers in the exploration module of RUN-GNN. In this section, we conduct an evaluation of the performance of PRGNN-based methods by varying the maximum subgraph size.

Figure 5 illustrates that our proposed RUN-GNN model achieves the best performance when the maximum subgraph size used for inference is greater than 4 hops. This result suggests that our method is particularly efficient at encoding and exploiting long relational paths. Furthermore, the advantage achieved by RUN-GNN over RED-GNN does not decrease significantly as the subgraph size grows. This finding indicates that our proposed strategy of utilizing *buffering update mechanism* to enhance relational rules cannot be replaced by simply increasing the number $n$ of G-GAT Layers in the exploration module.

## D  The Detailed Time Complexity Analysis

The overall time complexity of RUN-GNN, which consists of $l$ GNN layers in the exploration module and $n$ GNN layers in the buffer module, can be expressed as $O((\min(nD^l, (l + n)|\mathcal{F}|) + |\mathcal{V}|)d^2)$, where $d$ is the dimension of relation representations in the model, $D = \frac{|\mathcal{F}|}{|\mathcal{V}|}$ is the average degree of the knowledge graph , $|\mathcal{V}|$ is the number of entities, and $|\mathcal{F}|$ is the number of fact triples.

The time complexity comparison of these models is shown in the table 8. Below is a detailed analysis of the time complexity of RUN-GNN and other PRGNN methods.

### D.1  Theoretical Analysis

Let $|\mathcal{V}|$ be the number of entities and $|\mathcal{F}|$ be the number of fact triples. Similar to the NBF-Net(Zhu et al., 2021), we decompose the time complexity of the model into the MESSAGE and AGGREGATE parts.

#### D.1.1  Time Complexity of NBF-Net

The time complexity of the MESSAGE function in NBF-Net is $O(d)$, and the time complexity of the AGGREGATE function is $O(d^2)$. Since each information propagation is performed on the entire knowledge graph, the overall time complexity is $O(l(|\mathcal{F}|d + |\mathcal{V}|d^2))$.

#### D.1.2  Time Complexity of REDGNN

The time complexity of the MESSAGE function in REDGNN(Zhang and Yao, 2022) is $O(d)$, and the time complexity of the AGGREGATE function is $O(d^2)$. Therefore, the overall time complexity of REDGNN is $O(\min(D^l, l|\mathcal{F}|)d + |\mathcal{V}|d^2)$.

#### D.1.3  Time Complexity of RUN-GNN

RUN-GNN also performs progressive information propagation by sampling a series of subgraphs centered around the query entity, from small to large, and performing message passing on these subgraphs sequentially. The buffer update mechanism in RUN-GNN does not involve additional subgraph sampling after message passing in the GNNs of all exploration module. Instead, it continues to perform message passing on the largest subgraph used in the current reasoning process. Therefore, this step does not incur additional time cost for subgraph sampling.

Its MESSAGE function is QRFGU, which is essentially a variant of GRU function with additional attention, with a time complexity of $O(d^2)$. The AGGREGATE function is a simple linear transformation with a time complexity of $O(d^2)$. Therefore, the overall time complexity of RUN-GNN with $l$ GNN layers in the exploration module and $n$ GNN layers in the buffer module is $O((\min(nD^l, (l + n)|\mathcal{F}|) + |\mathcal{V}|)d^2)$.

| Methods | NBF-Net | REDGNN | RUN-GNN |
|---|---|---|---|
| Subgraph sample | no | yes | yes |
| MESSAGE | $O(d)$ | $O(d)$ | $O(d^2)$ |
| AGGREGRATE | $O(d^2)$ | $O(d^2)$ | $O(d^2)$ |
| Basic layer | $O(l(\|\mathcal{F}\|d + \|\mathcal{V}\|d^2))$ | $O(min(D^l, l\|\mathcal{F}\|)d + \|\mathcal{V}\|d^2)$ | $O(min(D^l, l\|\mathcal{F}\|)d^2 + \|\mathcal{V}\|d^2)$ |
| Buffer update | no | no | $min(D^l, \|\mathcal{F}\|)nd^2$ |
| Total | $O(l(\|\mathcal{F}\|d + \|\mathcal{V}\|d^2))$ | $O(min(D^l, l\|\mathcal{F}\|)d + \|\mathcal{V}\|d^2)$ | $O((min(nD^l, (l+n)\|\mathcal{F}\|) + \|\mathcal{V}\|)d^2)$ |

Table 8: Time complexity comparison table for different PRGNN methods.

| Methods | $l$ | $n$ | Time cost per epoch | MRR |
|---|---|---|---|---|
| NBF-Net | 5 | 0 | 3840 | 0.5439 |
| REDGNN | 5 | 0 | 218 | 0.5447 |
| REDGNN | 8 | 0 | 4457 | 0.5646 |
| RUN-GNN | 5 | 0 | 460 | 0.5636 |
| RUN-GNN | 5 | 3 | 508 | 0.5703 |

Table 9: Time comparison table for different PRGNN methods trained on the WN18RR dataset. Here, $l$ represents the number of GNN layers in the Exploration Module, and $n$ represents the number of GNN layers in the Buffer Module.

## D.2 Time Measurement and Empirical Analysis

As shown in the table 9, NBF-Net has a much higher training time for 1 epoch when the number of GNN layers in the exploration module is 5, as it performs message passing directly on the entire KG.

When the number of GNN layers in the buffer module is 0, i.e., without using the buffer update mechanism, the model's performance already surpasses that of REDGNN and NBF-Net with 5 GNN layers in the exploration module. The performance of RUN-GNN even approaches that of REDGNN with 8 GNN layers, which takes 8 times longer to train.

When the number of GNN layers in the buffer module is 3, i.e., using the buffer update mechanism, the model's reasoning ability is further improved with only a $\frac{1}{9}$ increase in time cost. It clearly outperforms REDGNN with 8 GNN layers, which requires more computational resources.

## E Case Study

In addition to its improved performance, our method RUN-GNN offers excellent interpretability. To illustrate this, we visualize the r-digraph utilized in the inference process for three queries from the family dataset, following the approach outlined in paper(Zhang and Yao, 2022). Figure 6 showcases the evidence discovered by our method during the inference process, which appears to be

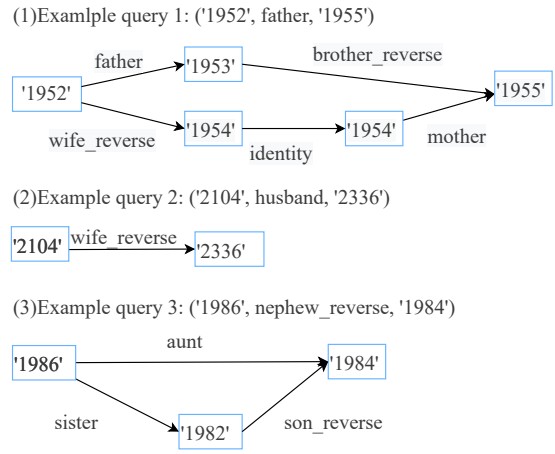

Figure 6: Examples of relational paths that are important when RUN-GNN performs inference.

logical and supports the notion that our approach can successfully learn crucial relational rules. This visualization further reinforces the interpretability and reliability of our method.