# OpenReview forum: "Towards Enhancing Relational Rules for Knowledge Graph Link Prediction"
_EMNLP/2023/Conference — EMNLP 2023 Findings_

### Official Review · Reviewer_55BH · 2023-07-31

**Soundness:** 3

**Excitement:**

3: Ambivalent: It has merits (e.g., it reports state-of-the-art results, the idea is nice), but there are key weaknesses (e.g., it describes incremental work), and it can significantly benefit from another round of revision. However, I won't object to accepting it if my co-reviewers champion it.

**Missing References:**

KG-based embedding methods and those utilizing non-Euclidean geometric spaces to perform learning updates have also recently been proposed:
- KDD 2019: Universal Representation Learning of Knowledge Bases by Jointly Embedding Instances and Ontological Concepts. In Proceedings of the 25th ACM SIGKDD International Conference on Knowledge Discovery &amp; Data Mining (KDD '19). Association for Computing Machinery, New York, NY, USA, 1709–1719. https://doi.org/10.1145/3292500.3330838
- KDD 2022: Dual-Geometric Space Embedding Model for Two-View Knowledge Graphs. In Proceedings of the 28th ACM SIGKDD Conference on Knowledge Discovery and Data Mining (KDD '22). Association for Computing Machinery, New York, NY, USA, 676–686. https://doi.org/10.1145/3534678.3539350

Furthermore, SOTA GNN models for large-scale multi-relational heterogeneous graphs using hierarchical attention mechanisms have also been proposed, which are missing:
- WWW 2019: Heterogeneous Graph Attention Network. In The World Wide Web Conference (WWW '19). Association for Computing Machinery, New York, NY, USA, 2022–2032. https://doi.org/10.1145/3308558.3313562
- ICDM 2021: "Bi-Level Attention Graph Neural Networks," 2021 IEEE International Conference on Data Mining (ICDM), Auckland, New Zealand, 2021, pp. 1126-1131, doi: 10.1109/ICDM51629.2021.00133.


**Paper Topic And Main Contributions:**

This paper proposes a knowledge graph reasoning approach, the Relational rUle eNhanced Graph Neural Network (RUN-GNN). RUN-GNN uses a query related fusion gate unit to model the sequentiality of relation composition and uses buffering update to help improve lagged entity information propagation. The authors perform experiments on various KG datasets for link prediction tasks.

**Questions For The Authors:**

- Can you more clearly justify the design choice for the components of your architecture e.g., reason for choosing G-GAT layers etc.
- What is the time complexity of your proposed approach?

**Reasons To Accept:**

- This paper generally motivates the problem of exploration, though some important references are missing in the Related Works section. The contributions of the work are concisely summarized.
- Architectural visualizations (particularly Figs. 2 & 3) are resourceful and follow well with the accompanied text.
- The authors use a variety of standard benchmark datasets for KGs. Experiment results also show good performance.
- Ablation studies visualizing and comparing the relation combination representation are insightful.


**Reasons To Reject:**

- In general, I have a concern regarding the novelty of the work, as it appears to simply be a combination of already proposed methods e.g., G-GAT layers, MLP etc.
- The relation works section is incomplete. Further, regarding the problem definition methodology, the authors also need to describe which view of the knowledge graph is part of the assumption e.g., instance-view, ontology-view, two-view KG etc.

**Reproducibility:**

2: Would be hard pressed to reproduce the results. The contribution depends on data that are simply not available outside the author's institution or consortium; not enough details are provided.

**Reviewer Confidence:**

3: Pretty sure, but there's a chance I missed something. Although I have a good feel for this area in general, I did not carefully check the paper's details, e.g., the math, experimental design, or novelty.

---

> ### Author Rebuttal · Authors · 2023-08-29
>
> Thank you for your insightful comments. We will address each of your concerns and clarify any areas where you have doubts.
>
> ### Q1. In general, I have a concern regarding the novelty of the work, as it appears to simply be a combination of already proposed methods e.g., G-GAT layers, MLP etc.
>
> A1:
>
> We appreciate your valuable comments.
>
> First and foremost, we need to clarify that **the G-GAT layer is an original component designed to enable the model to capture the sequentiality of compositional relations.**
>
> It is important to note that although we also employ attention mechanisms in graph neural networks, our G-GAT differs from GAT in the following aspects:
>
> 1. **Different application scenarios**: GAT is used in regular graphs, while our G-GAT is specifically designed for knowledge graphs. As G-GAT is applied to knowledge graphs, it requires the use of QRFGU to combine the head entities and relationships of the edges, which is not necessary in GAT.
>
> 2. **Different factors in computing attention**: GAT computes attention values based on the head and tail entities of the edges, without considering the relationships on the edges. On the other hand, our G-GAT only considers the messages to be propagated on the edges and the query relationship for the entire reasoning process, without considering the tail entities.
>
> By incorporating the fusion of the representations of the head entities and relationships **using G-GAT, the model is able to better capture the sequentiality of compositional relations and even more complex properties of relational rules.**
>
> On the other hand, we believe that the innovation of RUN-GNN lies not in the assembly of components, but the followings:
>
> During the analysis of PRGNN's operation, we identified two properties that affect the reasoning ability of PRGNN.
>
> **(1) The sequentiality of relation composition can impact the model's ability to correctly encode semantic information about relational rules.**
>
> **(2)  The lagged entity information propagation makes it difficult for the model to identify the correct relational rules.**
>
> However, existing PRGNN methods overlook these two properties. To address these issues, **we propose QRFGU and apply the G-GAT layer to encode relational rules**, which allows for modeling the sequentiality of relation composition. Additionally, **we introduce a buffering update mechanism to tackle the lagged entity information propagation.**
>
> ---
>
> ### Q2. The relation works section is incomplete. Further, regarding the problem definition methodology, the authors also need to describe which view of the knowledge graph is part of the assumption e.g., instance-view, ontology-view, two-view KG etc.
>
> A2:
>
> Thank you for your valuable suggestion and comments.
>
> Firstly, our approach is a method that learns and applies relational rules from factual triples in a knowledge graph using neural networks for link prediction. Therefore, it falls under the category of **instance-view methods**. We will clarify this point in the subsequent revised version of the paper.
>
> Secondly, in the related work, **we have followed the mainstream practices in knowledge graph link prediction and categorized previous methods based on how they utilize information from the knowledge graph.** Similar approaches have been adopted in existing studies [1], [2] to classify related work. Furthermore, research on knowledge graph link prediction presents related work from various perspectives based on the thematic needs of the paper. For example, paper [3] categorizes related work into Translational Distance Models, Semantic Matching Models, and GNN-based Models. Paper [4] categorizes related work into Transductive relation prediction and Inductive relation prediction.
>
> [1] Zhu Z, Zhang Z, Xhonneux L P, et al. Neural bellman-ford networks: A general graph neural network framework for link prediction[J]. Advances in Neural Information Processing Systems, 2021, 34: 29476-29490.
>
> [2] Zhang Y, Yao Q. Knowledge graph reasoning with relational digraph[C]//Proceedings of the ACM web conference 2022. 2022: 912-924.
>
> [3] Li R, Cao Y, Zhu Q, et al. How does knowledge graph embedding extrapolate to unseen data: a semantic evidence view[C]//Proceedings of the AAAI Conference on Artificial Intelligence. 2022, 36(5): 5781-5791.
>
> [4] Mai S, Zheng S, Yang Y, et al. Communicative message passing for inductive relation reasoning[C]//Proceedings of the AAAI Conference on Artificial Intelligence. 2021, 35(5): 4294-4302.
>
> ---
>
> ### Q3. Can you more clearly justify the design choice for the components of your architecture e.g., reason for choosing G-GAT layers etc.
>
> A3:
>
> Many thanks.
>
> We will clarify the design principles of our method and the factors considered when selecting components in the revised version. Below, we will rephrase some of our considerations in model structure and component design.
>
> Methods based on PRGNN, such as NBF-Net  and REDGNN, derive their reasoning capabilities from encoding relational rules in the model.
>
> In the aforementioned process, **ensuring that the model can encode valid relational rules and correctly encode the semantics of relational rules becomes crucial.** During the analysis of PRGNN's operation, we identified two properties that affect the reasoning ability of PRGNN.
>
> **(1) The sequentiality of relation composition can impact the model's ability to correctly encode semantic information about relational rules.**
>
> **(2)  The lagged entity information propagation makes it difficult for the model to identify the correct relational rules.**
>
> However, existing PRGNN methods overlook these two properties. To address these issues, **we propose QRFGU and apply the G-GAT layer to encode relational rules**, which allows for modeling the sequentiality of relation composition. Additionally, **we introduce a buffering update mechanism to tackle the lagged entity information propagation.**
>
> #### 3.1 QRFGU
>
> The QRFGU aims to **address the issue of existing PRGNN methods being unable to model the sequentiality of relation composition** We consider multiple factors when designing QRFGU.
>
> 1. Encoding the semantics of relation rules correctly helps the model perform accurate reasoning. The sequentiality of relation composition indicates that the order of relation composition determines the semantics of relation rules. Therefore, PRGNN requires a MESSAGE function that can model the sequentiality of relation composition.
>
> 2. RNN is a classic neural network widely used for sequence modeling, capable of effectively capturing properties such as sequence order.
>
> 3. The reasoning ability of PRGNN comes from selecting and encoding relation rules from relational paths. Path-based reasoning methods, such as RNNLogic [1], often use RNN to encode relational paths.
>
> 4. GRU in RNN includes various gates similar to attention mechanisms and a forget mechanism similar to residual connections, making it suitable for modeling complex sequences.
>
> Therefore, **we chose GRU [2] as the basic structure of QRFGU, ensuring that QRFGU can effectively model the sequentiality.**
>
> Furthermore, considering the importance of the query relation in relation rule selection and encoding, we **incorporate the query relation into the computation process of the forget gate and update gate, allowing the model to adjust the encoding of relation rule representation based on the query relation.**
>
> [1] Qu M, Chen J, Xhonneux L P, et al. RNNLogic: Learning Logic Rules for Reasoning on Knowledge Graphs[C]//International Conference on Learning Representations. 2020.
>
> [2] Chung J, Gulcehre C, Cho K, et al. Empirical evaluation of gated recurrent neural networks on sequence modeling[C]//NIPS 2014 Workshop on Deep Learning, December 2014. 2014.
>
> #### 3.2 G-GAT layer
>
> We refer to the GNN layer proposed and used in RUN-GNN as the Gated Graph Attention Network (G-GAT) layer.
>
> The G-GAT layer is the fundamental unit of RUN-GNN and consists of the following components:
>
> 1. The MESSAGE function combines the relation rule representation of the head entity in the triple with the representation of the relation to generate the candidate rule representation for the tail entity. Unlike existing methods like CompGCN [1, 2, 3] that use triple-based scoring functions, we use QRFGU, which can model the sequentiality and even more complex properties of relation composition.
>
> 2. The AGGREGATE function selects the most important candidate rule representations from multiple candidate rule representations corresponding to the same tail entity and generates the rule representation for the tail entity. Considering the influence of the query relation on the applied relation rules during reasoning, we employ a graph attention mechanism related to the query relation [4].
>
> [1] Vashishth S, Sanyal S, Nitin V, et al. Composition-based Multi-Relational Graph Convolutional Networks[C]//International Conference on Learning Representations. 2019.
>
> [2] Zhu Z, Zhang Z, Xhonneux L P, et al. Neural bellman-ford networks: A general graph neural network framework for link prediction[J]. Advances in Neural Information Processing Systems, 2021, 34: 29476-29490.
>
> [3] Zhang Y, Yao Q. Knowledge graph reasoning with relational digraph[C]//Proceedings of the ACM web conference 2022. 2022: 912-924.
>
> [4] Veličković P, Cucurull G, Casanova A, et al. Graph attention networks[J]. arXiv preprint arXiv:1710.10903, 2017.
>
>
>
> In addition, we have validated the effectiveness of the model structure and components in the ablation experiments of section 4.3.1. Furthermore, in sections 4.3.2 and 4.3.3, we experimentally have verified the effectiveness of our proposed method in addressing the two challenges.6
>
> ---
>
> ### Q4. What is the time complexity of your proposed approach?
>
> A4:
>
> Many thanks.
>
> The **overall time complexity of RUN-GNN**, which consists of $L$ GNN layers in the exploration modules and $n$ GNN layers in the buffer modules, can be **expressed as** $O((\min(nD^L,(L+n)|\mathcal{F}|) +|\mathcal{V}|)d^2)$, where $d$ is the dimension of relation representations in the model, $D$ is the average degree of the knowledge graph $\frac{|\mathcal{F}|}{|\mathcal{V}|}$, $|\mathcal{V}|$ is the number of entities, and $|\mathcal{F}|$ is the number of fact triples. We will add a description and analysis of the model's time complexity in the appendix in future revisions.
>
> Below is a detailed analysis of the time complexity of RUN-GNN and other PRGNN methods.
>
> #### 4.1 Theoretical Analysis
>
> Let $|\mathcal{V}|$ be the number of entities and $|\mathcal{F}|$ be the number of fact triples. Similar to the analysis in the NBF-Net paper, we decompose the time complexity of the model into the MESSAGE and AGGREGATE parts.
>
> ##### 4.1.2 Time Complexity of NBF-Net
>
> The time complexity of the MESSAGE function in NBF-Net is $O(d)$, and the time complexity of the AGGREGATE function is $O(d^2)$. Since each information propagation is performed on the entire knowledge graph, the overall time complexity is $O(L(|\mathcal{F}|d+|\mathcal{V}|d^2))$.
>
> ##### 4.1.3 Time Complexity of REDGNN
>
> The time complexity of the MESSAGE function in REDGNN is $O(d)$, and the time complexity of the AGGREGATE function is $O(d^2)$. Therefore, the overall time complexity of REDGNN is $O(\min(D^L,L|\mathcal{F}|) d+|\mathcal{V}|d^2)$.
>
> ##### 4.1.4 Time Complexity of RUN-GNN
>
> RUN-GNN also performs progressive information propagation by sampling a series of subgraphs centered around the query entity, from small to large, and performing message passing on these subgraphs sequentially. The buffer update mechanism in RUN-GNN does not involve additional subgraph sampling after message passing in the GNNs of all exploration modules. Instead, it continues to perform message passing on the largest subgraph used in the current reasoning process. Therefore, this step does not incur additional time cost for subgraph sampling.
>
> Its MESSAGE function is QRFGU, which is essentially a variant of GRU function with additional attention, with a time complexity of $O(d^2)$. The AGGREGATE function is a simple linear transformation with a time complexity of $O(d^2)$. Therefore, the overall time complexity of RUN-GNN with L GNN layers in the exploration modules and n GNN layers in the buffer modules is $O((\min(nD^L,(L+n)|\mathcal{F}|) +|\mathcal{V}|)d^2)$.
>
> Based on the analysis above, the time complexity comparison of these models is shown in the following table.
>
> | Methods | Subgraph sample | MESSAGE  | AGGREGRATE | Basic layer                                           | Buffer update                  | Total                                                     |
> | ------- | --------------- | -------- | ---------- | ----------------------------------------------------- | ------------------------------ | --------------------------------------------------------- |
> | NBF-Net | no              | $O(d)$   | $O(d^2)$   | $O(L(\|\mathcal{F}\|d+\|\mathcal{V}\|d^2))$           | no                             | $O(L(\|\mathcal{F}\|d+\|\mathcal{V}\|d^2))$               |
> | REDGNN  | yes             | $O(d)$   | $O(d^2)$   | $O(min(D^L,L\|\mathcal{F}\|) d+\|\mathcal{V}\|d^2)$   | no                             | $O(min(D^L,L\|\mathcal{F}\|) d+\|\mathcal{V}\|d^2)$       |
> | RUN-GNN | yes             | $O(d^2)$ | $O(d^2)$   | $O(min(D^L,L\|\mathcal{F}\|) d^2+\|\mathcal{V}\|d^2)$ | $min(D^L,\|\mathcal{F}\|)nd^2$ | $O((min(nD^L,(L+n)\|\mathcal{F}\|) +\|\mathcal{V}\|)d^2)$ |
>
> #### 4.2 Time Measurement and Empirical Analysis
>
> | Methods     | GNN layers number in Exploration Module | GNN layers number in Buffer Module | Time cost/per epoch | MRR        |
> | ----------- | --------------------------------------- | ---------------------------------- | ------------------- | ---------- |
> | NBF-Net     | 5                                       | 0                                  | 3840                | 0.5439     |
> | *REDGNN*    | *5*                                     | *0*                                | *218*               | *0.5447*   |
> | *REDGNN*    | *8*                                     | *0*                                | *4457*              | *0.5646*   |
> | **RUN-GNN** | **5**                                   | **0**                              | **460**             | **0.5636** |
> | **RUN-GNN** | **5**                                   | **3**                              | **508**             | **0.5703** |
>
> As shown in the table above, NBF-Net has a much higher training time for 1 epoch when the number of GNN layers in the exploration module is 5, as it performs message passing directly on the entire knowledge graph.
>
> When the number of GNN layers in the buffer module is 0, i.e., without using the buffer update mechanism, the model's performance already surpasses that of REDGNN and NBF-Net with 5 GNN layers in the exploration module. **The performance of RUN-GNN even approaches that of REDGNN with 8 GNN layers, which takes 8 times longer to train.**
>
> **When the number of GNN layers in the buffer module is 3, i.e., using the buffer update mechanism, the model's reasoning ability is further improved with only a $\frac{1}{9}$ increase in time cost. It clearly outperforms REDGNN with 8 GNN layers, which requires more computational resources.**
>
> ---
>
> ### Q5. Missing References
>
> A5:
>
> Thank you for your valuable suggestion. We will carefully reorganize the relevant sections on related work and incorporate the two types of methods [1,2,3,4] you mentioned.
>
> [1] Hao J, Chen M, Yu W, et al. Universal representation learning of knowledge bases by jointly embedding instances and ontological  concepts[C]//Proceedings of the 25th ACM SIGKDD International Conference on Knowledge Discovery & Data Mining. 2019: 1709-1719.
>
> [2] Iyer R G, Bai Y, Wang W, et al. Dual-geometric space embedding model for two-view knowledge graphs[C]//Proceedings of the 28th ACM SIGKDD Conference on Knowledge Discovery and Data Mining. 2022: 676-686.
>
> [3] Wang X, Ji H, Shi C, et al. Heterogeneous graph attention network[C]//The world wide web conference. 2019: 2022-2032.
>
> [4] Iyer R G, Wang W, Sun Y. Bi-level attention graph neural networks[C]//2021 IEEE International Conference on Data Mining (ICDM). IEEE, 2021: 1126-1131.
>
> ---
>
> To summarize, I sincerely appreciate your insightful comments. I hope this solution can address your issue and enhance the paper's overall quality.

---

### Official Review · Reviewer_G1kj · 2023-08-03

**Soundness:** 4

**Excitement:**

3: Ambivalent: It has merits (e.g., it reports state-of-the-art results, the idea is nice), but there are key weaknesses (e.g., it describes incremental work), and it can significantly benefit from another round of revision. However, I won't object to accepting it if my co-reviewers champion it.

**Missing References:**

Missing some inductive reasoning methods, such as: Inductive Relation Prediction with Logical Reasoning Using Contrastive Representations, Incorporating Context Graph with Logical Reasoning for Inductive Relation Prediction.

**Paper Topic And Main Contributions:**

Based on previous works, this paper aims to enhance the relational rules for knowledge graph link prediction. It addresses the issues of the sequentiality of relation composition and the lagged entity information propagation.

**Questions For The Authors:**

The reason to design Query Related Fusion Gate Unit. Is there any theoretical support?

**Reasons To Accept:**

(1) Based on RED-GNN and NBFNet, the paper proposes RUN-GNN to enhance relational rules, where a query related fusion gate unit and a buffering update mechanism are proposed.
(2) extensive experiments are carried out, including the transductive and inductive settings for link prediction. The results demonstrate the effectiveness of the proposed method.

**Reasons To Reject:**

(1) The writing of the paper needs to be improved. The paper aims to enhance relational rules in GNNs. But the motivation for it is not clearly stated. I suggest it should be improved in the introduction Section and display the learned rules in the experiment Section.
(2) The strategy of buffering update mechanism is to add extra GNN layers, which is simple and can significantly increase the calculation overhead of the model.
(3) There are missing some references, for example, some recent KG inductive methods.
(4) Figure 4 is confusing and hard to follow.


**Reproducibility:**

4: Could mostly reproduce the results, but there may be some variation because of sample variance or minor variations in their interpretation of the protocol or method.

**Reviewer Confidence:**

5: Positive that my evaluation is correct. I read the paper very carefully and I am very familiar with related work.

---

> ### Author Rebuttal · Authors · 2023-08-29
>
> Thank you for your insightful comments. We will address each of your concerns and clarify any areas where you have doubts.
>
> ### Q1. The writing of the paper needs to be improved. The paper aims to enhance relational rules in GNNs. But the motivation for it is not clearly stated. I suggest it should be improved in the introduction Section and display the learned rules in the experiment Section.
>
> A1:
>
> Thank you for your valuable suggestion. We will carefully review the paper and improve the wording in the revised version to enhance readability and ensure that the methods and theories are easily understandable. Due to the progressive propagation of relational information during the inference process, **the relational rules applied by RUN-GNN can be traced back through the attention values of each triplet calculated in the G-GAT computation process.** We will also showcase the learned rules in the revised paper.
>
>
>
> ### Q2. The strategy of buffering update mechanism is to add extra GNN layers, which is simple and can significantly increase the calculation overhead of the model.
>
> A2:
>
> Thank you for your valuable comment.
>
> **Since the additional GNN computations are performed only on existing subgraphs rather than the entire knowledge graph, the increased time consumption is not unacceptable compared to the overall time consumption.**
>
> Indeed, adding additional GNN layers with the buffer update mechanism does incur extra computational resource consumption. In previous PRGNN methods such as REDGNN, the computational resource consumption increases exponentially with the number of GNN layers because each additional GNN layer requires pre-expanding the subgraph. However, in the buffer update mechanism, we do not sample subgraphs, and the additional GNN layers only propagate information within the existing subgraphs, resulting in linearly increased computational resource consumption. Therefore, **we believe that the additional computational resource consumption is acceptable compared to the improvement in inference capability brought by the buffer update mechanism.**
>
> We will supplement the analysis of time complexity for RUN-GNN and other PRGNN methods in the appendix of the revised version. Below is a detailed analysis of the time complexity for RUN-GNN and other PRGNN methods.
>
> #### 2.1 Theoretical Analysis
>
> Let $|\mathcal{V}|$ denote the number of entities and $|\mathcal{F}|$ denote the number of fact triplets. Similar to the analysis in the NBF-Net paper, we decompose the time complexity of the model into two parts: MESSAGE and AGGREGATE.
>
> **The overall time complexity of RUN-GNN, which consists of $L$ GNN layers in the exploration modules and $n$ GNN layers in the buffer modules, can be expressed as $O((\min(nD^L,(L+n)|\mathcal{F}|) +|\mathcal{V}|)d^2)$,** where $d$ is the dimension of relation representations in the model, $D$ is the average degree of the knowledge graph $\frac{|\mathcal{F}|}{|\mathcal{V}|}$, $|\mathcal{V}|$ is the number of entities, and $|\mathcal{F}|$ is the number of fact triples.
>
> The time complexity comparison of these models is shown in the following table.
>
> | Methods | Subgraph sample | MESSAGE  | AGGREGRATE | Basic layer                                           | Buffer update                  | Total                                                     |
> | ------- | --------------- | -------- | ---------- | ----------------------------------------------------- | ------------------------------ | --------------------------------------------------------- |
> | NBF-Net | no              | $O(d)$   | $O(d^2)$   | $O(L(\|\mathcal{F}\|d+\|\mathcal{V}\|d^2))$           | no                             | $O(L(\|\mathcal{F}\|d+\|\mathcal{V}\|d^2))$               |
> | REDGNN  | yes             | $O(d)$   | $O(d^2)$   | $O(min(D^L,L\|\mathcal{F}\|) d+\|\mathcal{V}\|d^2)$   | no                             | $O(min(D^L,L\|\mathcal{F}\|) d+\|\mathcal{V}\|d^2)$       |
> | RUN-GNN | yes             | $O(d^2)$ | $O(d^2)$   | $O(min(D^L,L\|\mathcal{F}\|) d^2+\|\mathcal{V}\|d^2)$ | $min(D^L,\|\mathcal{F}\|)nd^2$ | $O((min(nD^L,(L+n)\|\mathcal{F}\|) +\|\mathcal{V}\|)d^2)$ |
>
> #### 2.2 Time Measurement and Empirical Analysis
>
> | Methods     | GNN layers number in Exploration Module | GNN layers number in Buffer Module | Time cost/per epoch | MRR        |
> | ----------- | --------------------------------------- | ---------------------------------- | ------------------- | ---------- |
> | NBF-Net     | 5                                       | 0                                  | 3840                | 0.5439     |
> | *REDGNN*    | *5*                                     | *0*                                | *218*               | *0.5447*   |
> | *REDGNN*    | *8*                                     | *0*                                | *4457*              | *0.5646*   |
> | **RUN-GNN** | **5**                                   | **0**                              | **460**             | **0.5636** |
> | **RUN-GNN** | **5**                                   | **3**                              | **508**             | **0.5703** |
>
> As shown in the table above, NBF-Net incurs much higher training time for 1 epoch when the number of GNN layers in the exploration module is 5, due to its direct message passing on the entire knowledge graph.
>
> When the number of GNN layers in the buffer module is 0, i.e., without using the buffer update mechanism, the model's performance already surpasses that of REDGNN and NBF-Net with 5 GNN layers in the exploration module. In fact, it even approaches the performance of REDGNN with 8 GNN layers, which takes 8 times longer to train.
>
> **When the number of GNN layers in the buffer module is 3, i.e., using the buffer update mechanism, the model's inference capability is further improved while only increasing the time cost by $\frac{1}{9}$. It clearly outperforms REDGNN with 8 GNN layers, which consumes more resources.**
>
> ###  Q3.There are missing some references, for example, some recent KG inductive methods.
>
> A3:
>
> Thank you for your valuable suggestion. We will incorporate more references in the revised version.
>
> The additional references we plan to include, but are not limited to, are [1, 2, 3, 4, 5, 6, 7].
> [1] Wang C, Zhou X, Pan S, et al. Exploring Relational Semantics for Inductive Knowledge Graph Completion[C]//Proceedings of the AAAI Conference on Artificial Intelligence. 2022, 36(4): 4184-4192.
>
> [2] Galkin M, Zhu Z, Ren H, et al. Inductive logical query answering in knowledge graphs[J]. Advances in Neural Information Processing Systems, 2022, 35: 15230-15243.
>
> [3] Xu X, Zhang P, He Y, et al. Subgraph neighboring relations infomax for inductive link prediction on knowledge graphs[J]. arXiv preprint arXiv:2208.00850, 2022.
>
> [4] Pan Y, Liu J, Zhang L, et al. Inductive relation prediction with logical reasoning using contrastive representations[C]//Proceedings of the 2022 Conference on Empirical Methods in Natural Language Processing. 2022: 4261-4274.
>
> [5] Lin Q, Liu J, Xu F, et al. Incorporating context graph with logical reasoning for inductive relation prediction[C]//Proceedings of the 45th International ACM SIGIR Conference on Research and Development in Information Retrieval. 2022: 893-903.
>
> [6] Mai S, Zheng S, Yang Y, et al. Communicative message passing for inductive relation reasoning[C]//Proceedings of the AAAI Conference on Artificial Intelligence. 2021, 35(5): 4294-4302.
>
> [7 ]Chen J, He H, Wu F, et al. Topology-aware correlations between relations for inductive link prediction in knowledge graphs[C]//Proceedings of the AAAI Conference on Artificial Intelligence. 2021, 35(7): 6271-6278.
>
> ###  Q4. Figure 4 is confusing and hard to follow.
>
> A4:
>
> We sincerely apologize for the confusion caused by the description of section 4.3.3 and Figure 4. We will carefully review the relevant content and optimize the experimental description to improve the readability of the paper.
>
> Section 4.3.3 aims to validate and demonstrate that our proposed QRFGU can effectively model the sequentiality of compositional relations in PRGNN.
>
> We select lots of common relational paths from the WN18RR dataset and revers the order of relations in these paths to obtain a set of reversed relational paths. For example, we reverse the order of relations in the path $father(x,y)\wedge daughter(y,z)$ to obtain $daughter(x,y)\wedge father(y,z)$. We encode the original relational paths and the reversed relational paths separately using the original RUN-GNN (represented as w/ QRFGU) and a variant using element-wise addition as the MESSAGE function (represented as w/o QRFGU). This result in four sets of representations. We map these representations to a 2-dimensional space and plot the representations of different sets of relational paths using different colored dots in Figure 4. We also connect the representations of the reversed relational paths and their corresponding original order relational paths using colored lines.
>
> **In Figure 4, there are no distinct orange lines, indicating that the representations of different order combinations of the same relation generated by the variant w/o QRFGU are very close.** The model fails to capture the sequentiality of compositional relations and cannot distinguish between these representations.
>
> On the other hand, **there are clear green lines in Figure 4, indicating that the representations of different order combinations of the same relation generated by RUN-GNN (w/ QRFGU) have significant differences.** The model successfully captures the sequentiality of compositional relations and can distinguish between these representations.
>
> The table below lists the symbols used in Figure 4 to represent the different sets of representations.
>
> |               | RUN-GNN（w/ QRFGU）    | Variant w/o QRFGU          |
> | ------------- | ---------------------- | -------------------------- |
> | Normal order  | Blue big  solid circle | Pink big  solid circle     |
> | Inverse order | Deep Blue star         | Purple small  solid circle |
>
> ### Q5. The reason to design Query Related Fusion Gate Unit. Is there any theoretical support?
>
> Many thanks. We will provide a clearer explanation of the theoretical support for the design of QRFGU in the subsequent revised version.
>
> The QRFGU aims to address the issue of existing PRGNN methods being unable to model the sequentiality of relation composition. In order for RUN-GNN to correctly encode the semantics of relation rules, we consider several factors when designing QRFGU.
>
> 1. Correctly encoding the semantics of relation rules helps the model perform accurate reasoning. The sequentiality of relation composition indicates that the order of relation combinations determines the semantics of relation rules. Therefore, PRGNN requires a MESSAGE function that can model the sequentiality of relation composition.
>
> 2. RNN is a classic neural network widely used for sequence modeling, which can effectively capture properties such as the sequentiality of sequences.
>
> 3. The reasoning ability of PRGNN comes from selecting and encoding relation rules from relational paths. Path-based reasoning methods, such as RNNLogic [3], also often use RNN to encode relational paths.
>
> 4. The GRU in RNN includes various gates similar to attention mechanisms and a forget mechanism similar to residual connections, which can effectively model complex sequences.
>
> Therefore, **we chose GRU [4] as the basic structure of QRFGU, ensuring that QRFGU can effectively model sequentiality.**
>
> Furthermore, considering the importance of query relations in relation rule selection and encoding, **we incorporate query relations into the computation process of the forget gate and update gate. This allows the model to adjust the encoding of relation rule representations based on the query relation.**
>
> In addition, we have validated the effectiveness of the QRFGU in the ablation experiments of section 4.3.1. Furthermore, in section 4.3.3, we experimentally have verified the effectiveness of QRFGU in modelling the sequentiality of relation composition.
>
> [1] Zhu Z, Zhang Z, Xhonneux L P, et al. Neural bellman-ford networks: A general graph neural network framework for link prediction[J]. Advances in Neural Information Processing Systems, 2021, 34: 29476-29490.
>
> [2] Zhang Y, Yao Q. Knowledge graph reasoning with relational digraph[C]//Proceedings of the ACM web conference 2022. 2022: 912-924.
>
> [3] Qu M, Chen J, Xhonneux L P, et al. RNNLogic: Learning Logic Rules for Reasoning on Knowledge Graphs[C]//International Conference on Learning Representations. 2020.
>
> [4] Chung J, Gulcehre C, Cho K, et al. Empirical evaluation of gated recurrent neural networks on sequence modeling[C]//NIPS 2014 Workshop on Deep Learning, December 2014. 2014.
>
> ### Q6. Missing some inductive reasoning methods, such as: Inductive Relation Prediction with Logical Reasoning Using Contrastive Representations, Incorporating Context Graph with Logical Reasoning for Inductive Relation Prediction.
>
> Thank you for your valuable feedback. We have conducted further research on the inductive reasoning method based on your suggestions and will supplement the missing references in the subsequent revised version to enhance the quality of the paper. The following are some of the literature sources [1, 2, 3, 4, 5, 6, 7] we have researched, which will be included in the references section of the revised version.
>
> [1] Wang C, Zhou X, Pan S, et al. Exploring Relational Semantics for Inductive Knowledge Graph Completion[C]//Proceedings of the AAAI Conference on Artificial Intelligence. 2022, 36(4): 4184-4192.
>
> [2] Galkin M, Zhu Z, Ren H, et al. Inductive logical query answering in knowledge graphs[J]. Advances in Neural Information Processing Systems, 2022, 35: 15230-15243.
>
> [3] Xu X, Zhang P, He Y, et al. Subgraph neighboring relations infomax for inductive link prediction on knowledge graphs[J]. arXiv preprint arXiv:2208.00850, 2022.
>
> [4] Pan Y, Liu J, Zhang L, et al. Inductive relation prediction with logical reasoning using contrastive representations[C]//Proceedings of the 2022 Conference on Empirical Methods in Natural Language Processing. 2022: 4261-4274.
>
> [5] Lin Q, Liu J, Xu F, et al. Incorporating context graph with logical reasoning for inductive relation prediction[C]//Proceedings of the 45th International ACM SIGIR Conference on Research and Development in Information Retrieval. 2022: 893-903.
>
> [6] Mai S, Zheng S, Yang Y, et al. Communicative message passing for inductive relation reasoning[C]//Proceedings of the AAAI Conference on Artificial Intelligence. 2021, 35(5): 4294-4302.
>
> [7 ]Chen J, He H, Wu F, et al. Topology-aware correlations between relations for inductive link prediction in knowledge graphs[C]//Proceedings of the AAAI Conference on Artificial Intelligence. 2021, 35(7): 6271-6278.
>
>
>
> To summarize, I sincerely appreciate your valuable suggestions and  comments. I trust that this solution addresses your concerns and enhances the paper's overall quality.

---

### Official Review · Reviewer_FhXe · 2023-08-11

**Soundness:** 3

**Excitement:**

4: Strong: This paper deepens the understanding of some phenomenon or lowers the barriers to an existing research direction.

**Paper Topic And Main Contributions:**

This paper proposes a revised model, RUN-GNN, based on PRGNN, which aims to leverage two attributes, (1) sequentially of relation composition, and (2) lagged entity information propagation. The main idea of RUN-GNN is to employ a query related fusion gate unit to model the first attribute, and buffering update mechanism for the second one.

**Questions For The Authors:**

See weakness.

**Reasons To Accept:**

1. Clearly presentation, especially for those nice figures.
2. The idea is interesting.

**Reasons To Reject:**

1. Why don’t you compare your model with baseline model, PRGNN? Besides, some of the important baselines should be compared, including [1] [2] [3] [4]. For more sota methods, refer to the survey papers [5, 6].

[1] D. Zhang, Z. Yuan, H. Liu, H. Xiong et al., “Learning to walk with dual agents for knowledge graph reasoning,” in Proc. of AAAI, 2022.

[2] Z. Zhang, J. Cai, Y. Zhang, and J. Wang, “Learning hierarchy-aware knowledge graph embeddings for link prediction,” in Proc. of AAAI, 2020.

[3] R. Li, J. Zhao, C. Li, D. He, Y. Wang, Y. Liu, H. Sun, S. Wang, W. Deng, Y. Shen et al., “House: Knowledge graph embedding with householder parameterization,” arXiv preprint arXiv:2202.07919, 2022.

[4] Y. Pan, J. Liu, L. Zhang, T. Zhao, Q. Lin, X. Hu, and Q. Wang, “Inductive relation prediction with logical reasoning using contrastive representations,” in Proceedings of the 2022 Conference on Empirical Methods in Natural Language Processing. Abu Dhabi, United Arab Emirates: Association for Computational Linguistics, Dec. 2022, pp. 4261–4274. [Online]. Available: https://aclanthology.org/2022.emnlp-main.286

[5] Liang K, Meng L, Liu M, et al. A Survey of Knowledge Graph Reasoning on Graph Types: Static, Dynamic, and Multimodal[J]. 2022.

[6] Wang J, Wang B, Qiu M, et al. A Survey on Temporal Knowledge Graph Completion: Taxonomy, Progress, and Prospects[J]. arXiv preprint arXiv:2308.02457, 2023.


2. More realistic and larger datasets should be considered, such as YAGO.
3. The subgraph extraction procedure may take lots of time. The running time should be reported and analysis.
4. More ablation studies should be included to illustrate the motivation of two proposed module. We can understand the motivation for why should we leverage the attributes. But can you explain how do you design frameworks of these modules? In other word, why do you come out the idea for the structure of the QRFGU in Figure 3. Why do you know it is effective?

**Reproducibility:**

4: Could mostly reproduce the results, but there may be some variation because of sample variance or minor variations in their interpretation of the protocol or method.

**Reviewer Confidence:**

4: Quite sure. I tried to check the important points carefully. It's unlikely, though conceivable, that I missed something that should affect my ratings.

---

> ### Author Rebuttal · Authors · 2023-08-29
>
> Thank you for your insightful comments. We will address your concerns and clarify any areas where you have doubts one by one.
>
> ### Q1. Why don’t you compare your model with baseline model, PRGNN? Besides, some of the important baselines should be compared, including [1] [2] [3] [4]. For more sota methods, refer to the survey papers [5, 6].
>
> A1:
>
> Thank you for your valuable suggestion.
>
> Firstly, we would like to clarify that **PRGNN is a generic term used to refer to a class of methods**, rather than a specific method. **The PRGNN includes NBF-Net and REDGNN. In the experimental section, we have compared our model with NBF-Net and REDGNN.**
>
> Secondly, we will make efforts to include as many baseline methods as possible in the revised version of the paper, as you have suggested. Here, we provide a partial comparison of the experimental results between the baselines you mentioned and our method:
>
> | Methods        |           | WN18RR   |          |           | FB25k-237 |          |
> | -------------- | --------- | -------- | -------- | --------- | --------- | -------- |
> |                | MRR       | Hit@1    | Hit@10   | MRR       | Hit@1     | Hit@10   |
> | CURL[1]        | 0.460     | 42.9     | 52.3     | 0.306     | 22.4      | 47.0     |
> | HAKE[2]        | 0.497     | 45.2     | 58.2     | 0.346     | 25.0      | 54.2     |
> | HousE[3]       | 0.511     | 46.5     | 60.2     | 0.361     | 26.6      | 55.1     |
> | RUN-GNN (ours) | **0.586** | **53.2** | **68.8** | **0.416** | **31.9**  | **61.0** |
>
> **It can be observed that our method still possesses significant advantages in reasoning capability compared to these baseline methods.**
>
> Among them, LogCo[4] is an inductive method that utilizes subgraph information. It requires collecting neighborhood subgraphs of all entities in the knowledge graph for each triple inference, resulting in a very high time complexity. Performing direct link prediction tasks with LogCo would require an extremely long time. We will continue to experiment with LogCo and include it in the baselines if possible.
>
> As for the baseline models HAKE and HousE, they have not been experimented on the Nell-995 dataset. CURL, on the other hand, uses a different NELL-995 dataset compared to the one we used. Reproducing the corresponding models and conducting experiments on our NELL-995 dataset will be done in our future work, and we will continue to supplement the experimental results accordingly.
>
> [1] Zhang D, Yuan Z, Liu H, et al. Learning to walk with dual agents for knowledge graph reasoning[C]//Proceedings of the AAAI Conference on Artificial Intelligence. 2022, 36(5): 5932-5941.
>
> [2] Zhang Z, Cai J, Zhang Y, et al. Learning hierarchy-aware knowledge graph embeddings for link prediction[C]//Proceedings of the AAAI conference on artificial intelligence. 2020, 34(03): 3065-3072.
>
> [3] Li R, Zhao J, Li C, et al. House: Knowledge graph embedding with householder parameterization[C]//International Conference on Machine Learning. PMLR, 2022: 13209-13224.
>
> [4] Pan Y, Liu J, Zhang L, et al. Inductive relation prediction with logical reasoning using contrastive representations[C]//Proceedings of the 2022 Conference on Empirical Methods in Natural Language Processing. 2022: 4261-4274.
>
> ---
>
> ### Q2. The subgraph extraction procedure may take lots of time. The running time should be reported and analysis.
>
> A2:
>
> Thank you for your valuable comment and suggestion. We will supplement the analysis of time complexity in the appendix of the revised version.
>
> Below, we will provide a brief analysis and comparison of the time complexity of RUN-GNN:
>
> #### 2.1 Theoretical Analysis
>
> **The overall time complexity of RUN-GNN, which consists of $L$ GNN layers in the exploration modules and $n$ GNN layers in the buffer modules, can be expressed as $O((\min(nD^L,(L+n)|\mathcal{F}|) +|\mathcal{V}|)d^2)$,** where $d$ is the dimension of relation representations in the model, $D= \frac{|\mathcal{F}|}{|\mathcal{V}|}$ is the average degree of the knowledge graph, $|\mathcal{V}|$ is the number of entities, and $|\mathcal{F}|$ is the number of fact triples.
>
> Below is a more detailed analysis of the time complexity:
>
> RUN-GNN  performs progressive information propagation by sampling a series of subgraphs centered around the queried entity and conducting message passing on these subgraphs in a sequential manner. The buffer update mechanism in RUN-GNN does not incur additional time cost for subgraph sampling after the message passing in the GNN layers of all exploration modules. Instead, it continues to perform message passing on the largest subgraph used in the current inference process.
>
> The MESSAGE function in RUN-GNN is QRFGU, which is essentially a variant of GRU function with additional attention. It has a time complexity of $O(d^2)$. The AGGREGATE function is a simple linear transformation with a time complexity of $O(d^2)$. Therefore, the overall time complexity of RUN-GNN with L exploration modules in the GNN layers and n buffer modules in the GNN layers is $O((min(nD^L,(L+n)|\mathcal{F}|) +|\mathcal{V}|)d^2)$.
>
> Based on the previous analysis, the time complexity comparison of these models is shown in the following table:
>
> | Methods | Subgraph sample | MESSAGE  | AGGREGRATE | Basic layer                                           | Buffer update                  | Total                                                     |
> | ------- | --------------- | -------- | ---------- | ----------------------------------------------------- | ------------------------------ | --------------------------------------------------------- |
> | NBF-Net | no              | $O(d)$   | $O(d^2)$   | $O(L(\|\mathcal{F}\|d+\|\mathcal{V}\|d^2))$           | no                             | $O(L(\|\mathcal{F}\|d+\|\mathcal{V}\|d^2))$               |
> | REDGNN  | yes             | $O(d)$   | $O(d^2)$   | $O(min(D^L,L\|\mathcal{F}\|) d+\|\mathcal{V}\|d^2)$   | no                             | $O(min(D^L,L\|\mathcal{F}\|) d+\|\mathcal{V}\|d^2)$       |
> | RUN-GNN | yes             | $O(d^2)$ | $O(d^2)$   | $O(min(D^L,L\|\mathcal{F}\|) d^2+\|\mathcal{V}\|d^2)$ | $min(D^L,\|\mathcal{F}\|)nd^2$ | $O((min(nD^L,(L+n)\|\mathcal{F}\|) +\|\mathcal{V}\|)d^2)$ |
>
> #### 2.2 Time Measurement and Empirical Analysis
>
> Table : time cost to train 1 epoch on WN18RR dataset with 1 Nvidia A4000 graphic card and final performance.
>
> | Methods     | GNN layers number in Exploration Module | GNN layers number in Buffer Module | Time cost/per epoch | MRR        |
> | ----------- | --------------------------------------- | ---------------------------------- | ------------------- | ---------- |
> | NBF-Net     | 5                                       | 0                                  | 3840s               | 0.5439     |
> | *REDGNN*    | *5*                                     | *0*                                | *218s*              | *0.5447*   |
> | *REDGNN*    | *8*                                     | *0*                                | *4457*s             | *0.5646*   |
> | **RUN-GNN** | **5**                                   | **0**                              | **460**s            | **0.5636** |
> | **RUN-GNN** | **5**                                   | **3**                              | **508s**            | **0.5703** |
>
>
>
> As shown in the table above, NBF-Net, which directly propagates messages throughout the entire knowledge graph, takes significantly more time to train for one epoch when the number of GNN layers in the exploration module is set to 5 compared to the other models.
>
> When the number of GNN layers in the buffer module is set to 0, indicating the absence of buffer update mechanism, the performance of the model already surpasses that of REDGNN and NBF-Net with 5 GNN layers in the exploration module. **In fact, it even approaches the performance of REDGNN with 8 GNN layers, which takes 8 times longer to train.**
>
> ---
>
> ### Q3. More realistic and larger datasets should be considered, such as YAGO.
>
> A3:
>
> Thank you for your valuable suggestion. Currently, our paper has utilized several classic datasets in the research of knowledge graph link prediction based on graph neural networks. Following your suggestion, we have presented the performance of our method on two more realistic datasets below. In the revised version of the paper, we will also supplement the performance of the model on these two datasets as well as on larger datasets.
>
> | Methods |           |  Family  |          |           |   UMLS   |          |
> | :-----: | :-------: | :------: | :------: | :-------: | :------: | :------: |
> |         |    MRR    |  Hit@1   |  Hit@10  |    MRR    |  Hit@1   |  Hit@10  |
> |  QuatE  |   0.941   |   89.6   |   99.1   |   0.944   |   90.5   |   99.3   |
> | MINERVA |   0.885   |   82.5   |   96.1   |   0.825   |   72.8   |   96.8   |
> |  DRUM   |   0.934   |   88.1   |   99.6   |   0.813   |   67.4   |  {97.6}  |
> | CompGCN |   0.933   |   88.3   |   99.1   |   0.927   |   86.7   |   99.4   |
> | REDGNN  | **0.992** | **98.8** | **99.7** |   0.964   |   94.6   |   99.0   |
> | RUN-GNN |   0.989   | **98.8** |   99.1   | **0.986** | **98.0** | **99.5** |
>
> ---
>
> ### Q4.More ablation studies should be included to illustrate the motivation of two proposed module. We can understand the motivation for why should we leverage the attributes. But can you explain how do you design frameworks of these modules? In other word, why do you come out the idea for the structure of the QRFGU in Figure 3. Why do you know it is effective?
>
> A4:
>
> Thank you for your valuable suggestions and insightful comments. We will carefully review the paper and improve the wording in the revised version to enhance readability and ensure that the methods and theories are easily understandable.
>
> In the following paragraph, we will restate our thought process behind the design of the model, aiming to address any concerns you may have.
>
> Methods based on PRGNN, such as NBF-Net and REDGNN, derive their reasoning capabilities from encoding relational rules in the model.
>
> In the aforementioned process, **ensuring that the model can encode valid relational rules and correctly encode the semantics of relational rules becomes crucial.** During the analysis of PRGNN's operation, we identified two properties that affect the reasoning ability of PRGNN.
>
> **(1) The sequentiality of relation composition can impact the model's ability to correctly encode semantic information about relational rules.**
>
> **(2)  The lagged entity information propagation makes it difficult for the model to identify the correct relational rules.**
>
> However, existing PRGNN methods overlook these two properties. To address these issues, **we propose QRFGU and apply the G-GAT layer to encode relational rules**, which allows for modeling the sequentiality of relation composition. Additionally, **we introduce a buffering update mechanism to tackle the lagged entity information propagation.**
>
> Below, we provide a more detailed description of the design motivations behind each component.
>
> #### 4.1 QRFGU
>
> The QRFGU aims to **address the issue of existing PRGNN methods being unable to model the sequentiality of relation composition**. We considered multiple factors when designing QRFGU.
>
> 1. Encoding the semantics of relational rules correctly helps the model perform accurate reasoning. The sequentiality of relation composition indicates that the order of relation combinations determines the semantics of relational rules. Therefore, PRGNN requires a MESSAGE function that can model the sequentiality of relation composition.
>
> 2. RNN is a classic neural network widely used for sequence modeling, capable of effectively capturing properties such as sequence order.
>
> 3. The reasoning capabilities of PRGNN stem from selecting and encoding relational rules from relational paths. Path-based reasoning methods, such as RNNLogic [3], also often use RNN to encode relational paths.
>
> 4. GRU in RNN incorporates various gates similar to attention mechanisms and a forget mechanism similar to residual connections, enabling it to model complex sequences effectively.
>
> Therefore, **we chose GRU [4] as the basic structure of QRFGU, ensuring that QRFGU can effectively model the sequentiality.** Furthermore, considering the significant role of the query relation in the selection and encoding of relational rules, **we incorporate the query relation into the computation process of the forget gate and update gate. This allows the model to adjust the encoding of relational rule representations based on the query relation.**
>
> [1] Zhu Z, Zhang Z, Xhonneux L P, et al. Neural bellman-ford networks: A general graph neural network framework for link prediction[J]. Advances in Neural Information Processing Systems, 2021, 34: 29476-29490.
>
> [2] Zhang Y, Yao Q. Knowledge graph reasoning with relational digraph[C]//Proceedings of the ACM web conference 2022. 2022: 912-924.
>
> [3] Qu M, Chen J, Xhonneux L P, et al. RNNLogic: Learning Logic Rules for Reasoning on Knowledge Graphs[C]//International Conference on Learning Representations. 2020.
>
> [4] Chung J, Gulcehre C, Cho K, et al. Empirical evaluation of gated recurrent neural networks on sequence modeling[C]//NIPS 2014 Workshop on Deep Learning, December 2014. 2014.
>
> #### 4.2 G-GAT layer
>
> We refer to the neural network layer that encodes messages using QRFGU for knowledge graph as the Gated Graph Attention Network (G-GAT) layer.
>
> The G-GAT layer is the fundamental unit of the RUN-GNN and consists of the following components:
>
> 1. The MESSAGE function combines the representation of the relation rule for the head entity in the triple and the representation of the relation to generate candidate rule representations for the tail entity. Unlike existing methods such as CompGCN [1, 2, 3] that use triplet-based scoring functions as MESSAGE, **we utilize QRFGU, which can model the sequentiality and even more complex properties of relation composition.**
>
> 2. The AGGREGATE function selects the most important candidate rule representations from multiple candidate rule representations corresponding to the same tail entity and generates the rule representation for the tail entity. Considering the influence of the query relation on the application of relation rules during reasoning, we incorporate a graph attention mechanism [4] that takes the query relation into account in this part.
>
> [1] Vashishth S, Sanyal S, Nitin V, et al. Composition-based Multi-Relational Graph Convolutional Networks[C]//International Conference on Learning Representations. 2019.
>
> [2] Zhu Z, Zhang Z, Xhonneux L P, et al. Neural bellman-ford networks: A general graph neural network framework for link prediction[J]. Advances in Neural Information Processing Systems, 2021, 34: 29476-29490.
>
> [3] Zhang Y, Yao Q. Knowledge graph reasoning with relational digraph[C]//Proceedings of the ACM web conference 2022. 2022: 912-924.
>
> [4] Veličković P, Cucurull G, Casanova A, et al. Graph attention networks[J]. arXiv preprint arXiv:1710.10903, 2017.
>
>
>
> In addition, we have validated the effectiveness of the model structure and components in the ablation experiments of section 4.3.1. Furthermore, in sections 4.3.2 and 4.3.3, we experimentally have verified the effectiveness of our proposed method in addressing the two challenges.
>
> ---
>
> In conclusion, thanks a lot for your comments. I hope this solves your problem and improves the quality of the paper.

---

### Meta-Review · Area_Chair_Ypmu · 2023-09-20

**Recommendation:** 3

**Metareview:**

This paper is built on top of the recent progressive relational graph neural network (PRGNN) methods for KG link prediction that has been proven to be quite effective. Specifically, this paper identifies two weaknesses of existing PRGNN methods and propose novel strategies to mitigate those. Experiments on multiple standard datasets with small KGs (WN18RR/FB15k-237/NELL-995) demonstrated the effectiveness of the proposed method, often setting a new state of the art on the datasets.

Strengths:
- The paper is well motivated and well executed. PRGNN is indeed a strong performing family of methods, and further solid improvement is meaningful.
- The writing and presentation generally make it easy to follow the logical flow and understand the main points
- The experiment results are impressive, considering that the performance on these datasets has largely saturated.

Weaknesses:
- As several reviewers pointed out, the paper was lacking an efficiency analysis, which is important considering that the proposed method adds new components. The authors are encouraged to incorporate that into the revised version.
- Evaluation on larger KGs such as YAGO and those in the Open Graph Benchmark would significantly strengthen the paper.
- The related work discussion is a bit insufficient for this specific topic (though the most related work seems to have been covered). The authors are encouraged to strengthen this part the revised version.

---

### Decision · Program_Chairs · 2023-10-07

**Decision:**

Accept-Findings

**Comment:**

This paper is built on top of the recent progressive relational graph neural network (PRGNN) methods for KG link prediction that has been proven to be quite effective. Specifically, this paper identifies two weaknesses of existing PRGNN methods and propose novel strategies to mitigate those. Experiments on multiple standard datasets with small KGs (WN18RR/FB15k-237/NELL-995) demonstrated the effectiveness of the proposed method, often setting a new state of the art on the datasets.

Strengths:
- The paper is well motivated and well executed. PRGNN is indeed a strong performing family of methods, and further solid improvement is meaningful.
- The writing and presentation generally make it easy to follow the logical flow and understand the main points
- The experiment results are impressive, considering that the performance on these datasets has largely saturated.

Weaknesses:
- As several reviewers pointed out, the paper was lacking an efficiency analysis, which is important considering that the proposed method adds new components. The authors are encouraged to incorporate that into the revised version.
- Evaluation on larger KGs such as YAGO and those in the Open Graph Benchmark would significantly strengthen the paper.
- The related work discussion is a bit insufficient for this specific topic (though the most related work seems to have been covered). The authors are encouraged to strengthen this part the revised version.